# LS-Merge: Merging Language Models in Latent Space

**Bedionita Soro**[1][*] **Aoxuan Silvia Zhang**[1][*] **Bruno Andreis**[3,4][*] **Jaehyeong Jo**[1]
**Song Chong**[1]  **Sung Ju Hwang**[1,2][†]
[1]KAIST  [2]DeepAuto.ai  [3]University of Oxford  [4]Slater Labs

## Abstract

Model merging in weight space is an efficient way to reuse pretrained models, but existing methods typically assume matching architectures or sizes, making heterogeneous merges brittle or infeasible. We address this limitation by encoding model weights into a smooth latent space, enabling cross-architecture operations, and performing the merge in the latent space before decoding back to weights. This approach faces two major challenges. First, LLMs contain billions of parameters, which makes latent encoding computationally demanding. Second, using high compression ratios often hinders the encoder's ability to generalize to unseen weights. We tackle these issues with a transformer-based variational autoencoder (VAE) trained in a two-stage compression curriculum with structured layer-aware chunking: the model first learns a high-capacity latent representation and then distills to a compact code, improving both stability and out-of-distribution generalization. To align heterogeneous models, we introduce a dimensionality-matching projection that allows interpolation between models of different sizes. Empirically, latent-space interpolation is consistently more robust than direct weight-space averaging and yields stronger downstream performance when merging models of different sizes. Together, these components provide a scalable, architecture-agnostic recipe for model merging.

## 1 Introduction

Large Language Models (LLMs) are foundational to modern artificial intelligence. However, their pretraining demands millions of GPU-hours, leading to a significant inefficiency if the acquired knowledge remains confined to a single model instance. To mitigate this cost and enhance utility, weight-space model merging has emerged as a promising approach. This technique combines parameters from multiple pretrained models to integrate complementary capabilities and improve performance on diverse tasks, all without additional pretraining (Yang et al., 2024).

Existing merging methodologies range from straightforward linear interpolation (Wortsman et al., 2022) to sophisticated evolutionary search-based fusion (Akiba et al., 2025), consistently demonstrating practical benefits at scale. Despite these advancements, current techniques typically face two significant limitations: (i) Requirement for multiple source models: Most approaches necessitate at least two distinct pretrained models for merging, which restricts their application when the goal is to enhance or adapt a single existing model. (ii) Architectural homogeneity: Merging methods frequently assume shape-compatible or architecturally homogeneous models, hindering their use with mismatched architectures (e.g., varying widths or depths) (Yu et al., 2024a). These constraints significantly limit the broad applicability of merging across diverse LLM checkpoints as well as self-merging. Addressing these constraints is crucial for unlocking the full potential of pretrained LLM checkpoints and fostering a more flexible and efficient paradigm for model reuse.

---

[*]Equal contribution.
[†]Correspondence to: Bedionita Soro `sorobedio@kaist.ac.kr`, Aoxuan Silvia Zhang `aoxuan_silvia_zhang@kaist.ac.kr`, Bruno Andreis `bruno.andreis@eng.ox.ac.uk`, Jaehyeong Jo `harryjo97@kaist.ac.kr`, Song Chong `songchong@kaist.ac.kr`, Sung Ju Hwang `sjhwang82@kaist.ac.kr`.

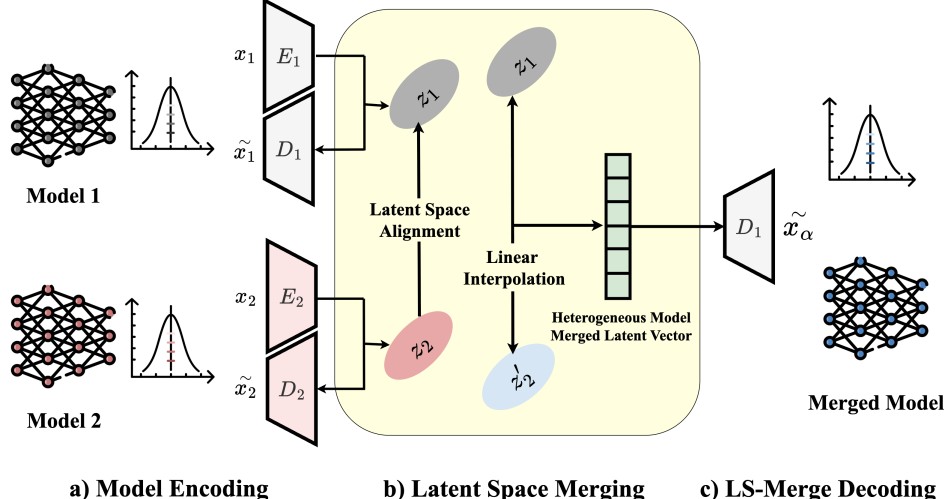

a) Model Encoding        b) Latent Space Merging        c) LS-Merge Decoding

Figure 1: **LS-Merge: encode, align, and decode for LLM weight merging. (a)** Pretrained weight tensors $x_1$ and $x_2$ are chunked layer-wise and encoded by $E_1, E_2$ into latents $z_1, z_2$; decoders $D_1, D_2$ reconstruct $\tilde{x}_1, \tilde{x}_2$. **(b)** A latent-space alignment places $z_1$ and $z_2$ in a shared manifold where they are merged(linear interpolation) yielding a merged latent $z_\alpha$, supporting both homogeneous (same-architecture(same encoding model) and heterogeneous (cross-architecture) merges. **(c)** A target decoder (e.g., $D_1$) maps $z_\alpha$ back to weights $\tilde{x}_\alpha$, producing the merged model. Here, $x_i$ denotes the full weight set of model $i$, $z_i$ its latent code, and tildes indicate reconstructions.

Recent advancements in weight-space learning (Sch"urholt et al., 2024; Schürholt et al., 2022; Peebles et al., 2022; Knyazev et al., 2025) offer a compelling opportunity to overcome the aforementioned limitations in LLM merging. These approaches operate directly on pretrained model weights as the input data modality. By projecting parameters from diverse models into a smooth, unified latent space of consistent dimensionality, we can inherently mitigate the challenge of architectural mismatch. Specifically, these parameter signals can be embedded into identical-dimensionality representations using powerful generative models such as Variational Autoencoders (VAEs) (Kingma & Welling, 2013), normalizing flows (Kobyzev et al., 2021), or diffusion models (Wang et al., 2025). While these innovative latent-space methods have demonstrated considerable success in vision and other domains, their application to the encoding and subsequent merging of pretrained LLM weights remains a largely unexplored frontier.

We propose LS-Merge, a novel framework that fundamentally shifts the merging process from the raw weight space to a learned latent space. This paradigm enables both homogeneous and heterogeneous merging by design: (i) Enabling single-model augmentation: A generative model can learn the latent manifold of a single LLM parameters, facilitating "merging" operations within this latent space. This allows for the exploration of variations, specializations, or the enhancement of capabilities derived solely from the original model, obviating the need for an external second model. (ii) Facilitating heterogeneous integration: By projecting diverse LLM architectures into a common, fixed-dimensional latent space, the stringent constraints of architectural homogeneity are significantly relaxed. Merging operations then occur within this unified latent representation, enabling the seamless integration of knowledge from models with differing widths, depths, or even fundamental architectural designs, as our method operates solely on the tensor data within the latent space. In summary, LS-Merge advances LLM merging through the following key contributions:

- **Weight statistics that matter for encoding.** We show that LLM weights exhibit low variance with heavy tails, that could significantly affect the choice of encoding network.

- **Merging LLMs in latent space.** We propose LS-Merge, a novel latent-space merging methodology that enables merging LLMs in their weights latent space.

- **Heterogeneous merging.** We introduce a dimensionality-matching projection and OT-based latent alignment that enables interpolation between models of different depths or widths.

- **Consistent empirical gains.** On Gemma and LLaMA pretrained models and LoRA-experts merging, our latent space merging method outperforms existing merging methods and remains robust under heterogeneity.

## 2 RELATED WORK

**Model averaging** The simplest merging approach directly averages weights. Model Soup (Wortsman et al., 2022) showed that averaging fine-tuned checkpoints from the same initialization improves generalization. Extensions include Uniform Soup (Wortsman et al., 2022) for LoRA weights and spherical linear interpolation (SLERP), which interpolates on the unit hypersphere but remains a pairwise-only method that merges only two models at a time and requires identical architectures.

**Interference-aware techniques** Merging models from different tasks can cause misalignment. Yadav et al. (2023) uses trimming and sign alignment, Yu et al. (2024a) sparsifies and rescales fine-tuned deltas, and Yu et al. (2024b) combine both. Greedy Soup (Wortsman et al., 2022) and EvolMerge (Akiba et al., 2025) treat merging as an optimization problem, while Task Arithmetic (Ilharco et al., 2023) models task vectors in parameter space. These methods require aligned architectures and incur computational cost.

**Modular assembly** Instead of merging weights, some approaches combine modules or experts. Model Stocks (Jang et al., 2024) and LoraHub (Huang et al., 2023) aligns LoRA adapters, Pack of LLMs (Mavromatis et al., 2024) learns expert routing, and cBTM (Gururangan et al., 2023) merges task-specific experts via unsupervised domain discovery. Modular systems are flexible but increase inference cost and do not unify knowledge into a single parameter set.

All prior methods operate in weight or module space and assume architectural alignment. In contrast, we propose **latent-space model merging**, which generalizes beyond these constraints.

## 3 LS MERGE

We present **LS-Merge**, a framework for performing model merging *in latent space*. As illustrated in Figure 1, we encode pretrained LLM weights, operate on their latent representations (including alignment when heterogeneous), and decode back to parameters to obtain the merged model.

### 3.1 EXPLORING WEIGHT DYNAMICS IN LLMs

To motivate our design, we analyze the distributional properties of transformer submodules in Gemma-3 (Team et al., 2025) and LLaMA-3.2 (Grattafiori et al., 2024). We compute the first four moments (mean, variance, skewness, and kurtosis) for self-attention layers (q_proj, k_proj, and o_proj) and MLP layers (up_proj, down_proj, and gate_proj). Table 1 reports the statics stands for per moment with full details in appendix Table 9. Weights exhibit near-zero means, low variances, and small positive skewness, but *markedly high kurtosis* in earlier layers, especially in self-attention projections. High kurtosis, i.e., leptokurtic distributions, indicates sharp peaks with heavy tails: rare, large-magnitude parameters that are likely functionally important. This contradicts Gaussian assumptions used in prior work (Si et al., 2025) and suggests that encoders must preserve tail events rather than over-regularize toward a narrow Gaussian. We observe this pattern consistently across sizes within both families.

**Theoretical Compressibility of LLM Weights** As the LLM weights distribution shows heavy tails and low intrinsic variance, we investigate whether a simple VAE can compress LLM weights effectively. Let $W \in \mathbb{R}^{n \times m}$ be a layer matrix and let $w = \text{vec}_{\text{row}}(W) \in \mathbb{R}^D$ denote the row-wise concatenation (flattening) of $W$ with $D = nm$. Empirically (Fig. 2), the top $r \ll \min(n, m)$ principal components capture nearly all variance, i.e., $\sum_{i=1}^{r} \lambda_i / \sum_{i=1}^{D} \lambda_i \approx 1$, where $\{\lambda_i\}$ are the eigenvalues of the empirical covariance of $w$ (equivalently, proportional to squared singular values of $W$). By Eckart–Young theorem (Eckart & Young, 1936), the best rank-$r$ approximation $W_r$ satisfies $\|W - W_r\|_F^2 = \sum_{i>r} \lambda_i$ and can be stored with $O(r(n + m))$ parameters (e.g., $U_r \in \mathbb{R}^{n \times r}$, $V_r \in \mathbb{R}^{m \times r}$). Thus, weights concentrate near a low-dimensional set. Assuming the collection of flattened weights $\mathcal{W} \subset \mathbb{R}^D$ concentrates near a smooth $d$-dimensional manifold $\mathcal{M}$ with $d \ll D$, manifold embedding results (Bengio et al., 2012; Lahiri et al., 2016) imply the existence of

| layers | llama3-2-3b-it | | | | gemma-3-1b-it | | | | gemma-3-4b-it | | | |
|---|---|---|---|---|---|---|---|---|---|---|---|---|
| | mean | var. | skew | kurt. | mean | var. | skew | kurt. | mean | var. | skew | kurt. |
| self_attn | 0 | 0.0017 | 0.0192 | 8.4032 | 0.0002 | 0.0031 | 0.0500 | 15.0505 | 0 | 0.0015 | 0.0501 | 15.2010 |
| | 0 | 0.0013 | 0.0191 | 7.3438 | 0.0002 | 0.0030 | 0.0450 | 9.8347 | 0 | 0.0015 | 0.0267 | 7.3832 |
| | 0 | 0.0012 | 0.0178 | 6.2172 | 0.0001 | 0.0030 | 0.0389 | 9.0363 | 0 | 0.0014 | 0.0255 | 7.2277 |
| | 0 | 0.0011 | 0.0104 | 5.4477 | 0.0001 | 0.0030 | 0.0288 | 8.6496 | 0 | 0.0014 | 0.0172 | 6.0731 |
| avg (self_attn) | 0 | 0.0005 | -0.0002 | 1.4342 | 0 | 0.0012 | -0.0009 | 3.2858 | 0 | 0.0005 | 0.0009 | 2.6900 |
| mlp | 0 | 0.0005 | 0.0093 | 5.4740 | 0.0002 | 0.0010 | 0.0167 | 8.7665 | 0 | 0.0004 | 0.0266 | 6.3297 |
| | 0 | 0.0005 | 0.0080 | 4.1465 | 0.0002 | 0.0010 | 0.0090 | 3.1514 | 0 | 0.0004 | 0.0155 | 5.9670 |
| | 0 | 0.0004 | 0.0075 | 2.6694 | 0.0002 | 0.0010 | 0.0077 | 3.0577 | 0 | 0.0004 | 0.0130 | 2.7454 |
| | 0 | 0.0004 | 0.0074 | 2.4364 | 0.0002 | 0.0010 | 0.0071 | 3.0113 | 0 | 0.0004 | 0.0094 | 2.4371 |
| avg (mlp) | 0 | 0.0003 | 0.0003 | 0.8435 | 0 | 0.0006 | 0 | 1.1739 | 0 | 0.0003 | 0.0006 | 1.0807 |

Table 1: Layer-wise distribution statistics of model parameters for three instruction-tuned LLMs: **llama3-2-3b-it**, **gemma-3-1b-it**, and **gemma-3-4b-it**. For each model and block type (`self_attn` and `mlp`), we report the first four moments of the flattened weight tensors: *mean*, *variance*, *skewness*, and *excess kurtosis*. Per-row entries list representative layers in each block; the `avg` row aggregates across all layers for that block and model. Means are near zero and variances are small, while skewness and especially kurtosis (often $> 5$ and up to $\sim 15$) indicate pronounced heavy tails and asymmetry. These non-Gaussian, heavy-tailed statistics motivate encoders that preserve rare but high-magnitude outliers.

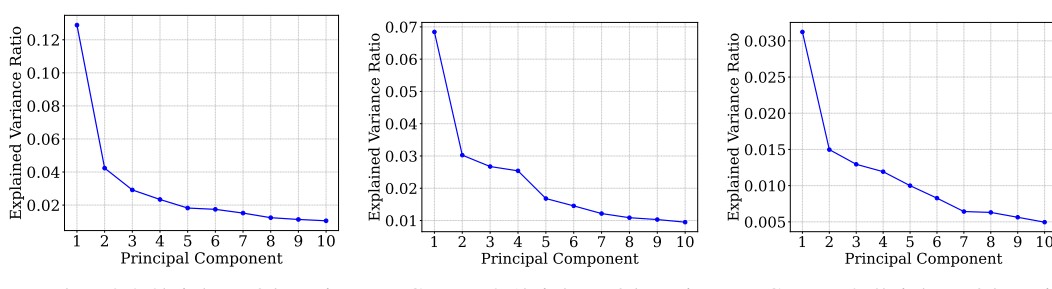

(a) Llama3-2-3b-it layer 0 k_proj   (b) Gemma-3-1b-it layer 0 k_proj   (c) Gemma-3-4b-it layer 0 k_proj

Figure 2: Plots of the PCA **explained-variance ratio** for first k projection weight matrices per model from three LLM checkpoints: Llama-3-2-3b-it, Gemma-3-1b-it, and Gemma-3-4b-it. Additional results are reported in the appendix A. For each model, we show the four projection matrices of self-attention (*q, k, v, o*). The sharp drop after the leading principal components highlights a low-rank structure that is consistent across architectures and model sizes.

$\Phi : \mathbb{R}^D \to \mathbb{R}^k$ with $k = O\left(\frac{d}{\varepsilon^2} \log \frac{V}{\tau}\right) \ll D$ that preserves pairwise distances on $\mathcal{M}$ within $(1 \pm \varepsilon)$, where $V$ bounds manifold volume and $\tau$ its reach. This confirms that there exists a projection map to a lower dimension which can be approximated by an encoder A VAE encoder $f_\theta : \mathbb{R}^D \to \mathbb{R}^k$ with decoder $g_\phi : \mathbb{R}^k \to \mathbb{R}^D$ can approximate such compressive embeddings while modeling uncertainty, justifying our latent compression.

## 3.2 LLM Weights Preprocessing and Encoding

We first standardize pretrained LLM weights of varying shapes and store the normalized tensors offline. We then learn a compact latent representation with a transformer-based VAE, chosen for its strong generalization to unseen checkpoints and faster training than convolutional alternatives at a comparable parameter count.

**Preprocessing as a Sequence** For each layer, we flatten its weight tensor to $w \in \mathbb{R}^L$, then zero-pad to $L_p = \lceil L/c \rceil c$ and partition into $n = L_p/c$ non-overlapping chunks $\{c_i\}_{i=1}^n$ of size $c$. A batch becomes $X \in \mathbb{R}^{B \times n \times c}$. Each chunk is embedded to $X_{\text{emb}} \in \mathbb{R}^{B \times n \times d}$ and passed through a transformer encoder $E_\theta$ with optional token downsampling. The latent is

$$z = E_\theta(X_{\text{emb}}) \in \begin{cases} \mathbb{R}^{B \times z_d} & \text{(pooled over tokens)} \\ \mathbb{R}^{B \times n \times z_d} & \text{(token-wise)} \end{cases}$$

The decoder $D_\phi$ is trained to reconstruct chunked weights from the latent $z$. We optimize the $\beta$-VAE objective in eq. 1

$$\mathcal{L} \;=\; -\,\mathbb{E}_{q_\phi(z\,|\,w)}\big[\log p_\theta(w\,|\,z)\big] \;+\; \beta\,\mathrm{KL}(q_\phi(z\,|\,w)\,\|\,p(z))\,, \tag{1}$$

with $p(z)$ standard Gaussian and fixed $\beta$.

**Stabilizing Training on Heavy-Tailed Weights**   Since LLM weights distribution has low variance and heavy tails (Section 3.1; Appendix 5), training VAEs on these weights can lead to collapse in the early training stage. We use (i) transformer blocks (Vaswani et al., 2017) in $E_\theta, D_\phi$ for long-range coupling across chunks and (ii) a two-stage curriculum: first train a deterministic autoencoder (KL off) to convergence, then enable the KL term and fine-tune to structure the latent space without sacrificing fidelity. The VAE performance is measured by the performance of the reconstructed weights when used to initialize the corresponding architectures.

### 3.3   LATENT SPACE ALIGNMENT AND MERGING

**Self-Merging and Homogeneous Merging**   *Self-merging* encodes a single model and draws multiple latent codes from its posterior (or the prior) to combine them, which is equivalent to merging *homogeneous* models whose per-layer embeddings share the same dimensionality. To be specific, for two checkpoints with weights $W_a, W_b \in \mathbb{R}^N$, we encode $z_a = E(W_a)$ and $z_b = E(W_b)$, and linearly interpolate them as $z_\lambda = (1 - \lambda)z_a + \lambda z_b$ for $\lambda \in [0, 1]$. We obtain the weights by decoding the interpolated latent $\widehat{W}_\lambda = D(z_\lambda)$. Empirically, latent-domain interpolation better preserves functional coherence than direct weight-space mixing, and common merge operators, for example, model soup or task arithmetic (Wortsman et al., 2022; Ilharco et al., 2023), transfer naturally by applying them to $\{z_a, z_b\}$ before decoding.

**Heterogeneous Mapping (depth/width mismatch)** When two architectures match, layer by layer, in the number of weight chunks, we employ a single VAE to embed all layers into a common $d$-dimensional latent space. If the per-layer number and chunk counts differ, we instead deploy separate encoders for each architecture. Let the source have $n_s$ layers with size $M$, and the target $n_t$ layers with size $N$. We embed per-layer latents to a fixed dimension $d$ and rescale the source so that total capacity matches the target:

$$r \;=\; \frac{n_t\,N}{n_s\,M}, \quad Z^{(\text{src, mapped})} \in \mathbb{R}^{n_t \times d}, \quad Z^{(\text{tgt})} \in \mathbb{R}^{n_t \times d}.$$

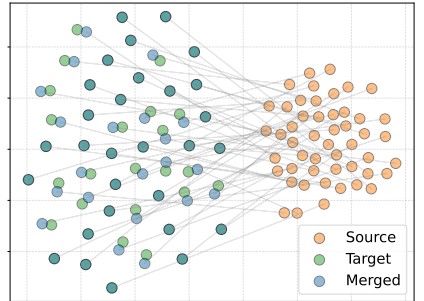

Figure 3: Visualization of embeddings of Gemma models and OT-aligned latents. The merged latent is partially overlapped with the target latent.

This proportional alignment across depth and width standardizes each architecture to a common per-layer latent dimensionality, enabling interpolation and the use of standard merge operators. However, when VAEs are trained separately per architecture (or under different training regimes), their *latent distributions* can differ, so equal shape does not guarantee comparability.

**Optimal Transport Alignment.**   While latent encoding standardizes per-layer dimensionality, it does not guarantee that two models' latent representations are geometrically compatible. As shown in Appendix C (Figure 9a), homogeneous models (e.g., checkpoints from the same pre-training run) exhibit overlapping latent support and often satisfy *Linear Mode Connectivity*, making linear interpolation well-behaved. In contrast, heterogeneous models such as Gemma and Llama produce latent distributions $\mathcal{Z}_{\text{src}}$, $\mathcal{Z}_{\text{tgt}}$ that lie on disjoint manifolds with different covariance structures and density profiles (Figure. 9b). Interpolating between such misaligned latents could produce low performing weights that fall outside the target decoder's valid manifold and degrade functionality(performance on down stream tasks). To address this, we treat heterogeneous merging as a *manifold registration* problem and align the source latent distribution to the target using Optimal Transport (OT) (Villani, 2009; Santambrogio, 2015). OT provides a principled framework for finding a map $T : \mathcal{Z}_{\text{src}} \rightarrow \mathcal{Z}_{\text{tgt}}$ that minimizes geometric distortion while reshaping one distribution into the

---

**Algorithm 1** Heterogeneous LLM Parameter Merging in Latent Space

---

**Require:** Source weights $W_{\text{src}}$, target weights $W_{\text{tgt}}$, encoder–decoder $(E, D)$
**Ensure:** Merged weights $W_{\text{merged}}$

1: Extract per-layer groups $L_{\text{src}}, L_{\text{tgt}}$; set $N \leftarrow \min(|L_{\text{src}}|, |L_{\text{tgt}}|)$ and define pairs $(l_{\text{src}}^{(j)}, l_{\text{tgt}}^{(j)})_{j=1}^{N}$
2: **for** $j = 1$ to $N$ **do**
3:    Flatten & chunk $w_{\text{src}}^{(j)}, w_{\text{tgt}}^{(j)}$; encode $z_{\text{src}}^{(j)} \leftarrow E(w_{\text{src}}^{(j)}), z_{\text{tgt}}^{(j)} \leftarrow E(w_{\text{tgt}}^{(j)})$
4:    Proportional mapping to fixed $d$: obtain $Z_{\text{src}}^{(j)}, Z_{\text{tgt}}^{(j)} \in \mathbb{R}^{n_t \times d}$
5:    OT alignment: $\tilde{Z}_{\text{src}}^{(j)} \leftarrow \text{OT-ALIGN}\left(Z_{\text{src}}^{(j)}, Z_{\text{tgt}}^{(j)}\right)$  (solve equation 2)
6:    Interpolate latents: $Z_{\lambda}^{(j)} \leftarrow (1 - \lambda) Z_{\text{tgt}}^{(j)} + \lambda \tilde{Z}_{\text{src}}^{(j)}$
7:    Decode & assemble: $w_{\text{merged}}^{(j)} \leftarrow D\left(Z_{\lambda}^{(j)}\right)$; place into $W_{\text{merged}}$
8: **end for**
9: **return** $W_{\text{merged}}$

---

other. Formally, we solve the Monge problem under the 2-Wasserstein distance 2:

$$T^* = \arg\min_{T} \int \|z - T(z)\|_2^2 \, d\mu_{\text{src}}(z) \quad \text{s.t.} \quad T_{\#}\mu_{\text{src}} = \mu_{\text{tgt}}, \tag{2}$$

where $T_{\#}\mu_{\text{src}}$ denotes the pushforward measure induced by $T$. This ensures that applying $T^*$ to source latents produces samples distributed as the target. While the general Monge problem is computationally intensive, it can approximate each layer's latent distribution as a high-dimensional Gaussian defined by its empirical mean $\mu_s$ and covariance $\Sigma_s$. Under this assumption, the optimal transport map admits a closed-form affine solution: $\tilde{z}_{\text{src}} = T^*(z_{\text{src}}) = \mu_t + A(z_{\text{src}} - \mu_s)$, where $A = \Sigma_s^{-1/2} \left(\Sigma_s^{1/2} \Sigma_t \Sigma_s^{1/2}\right)^{1/2} \Sigma_s^{-1/2}$. This transformation aligns both the mean and covariance of the source latents to the target, effectively registering the two manifolds and removing the geometric mismatch. In practice, we use existing OT library from Flamary et al. (2021; 2024) in our work.

After alignment, the transported latents $\tilde{Z}_{\text{src}}$ and the target latents $Z_{\text{tgt}}$ share a common support, and we perform interpolation in this aligned space: $Z_{\lambda}^{\text{OT}} = (1 - \lambda) Z_{\text{tgt}} + \lambda \tilde{Z}_{\text{src}}$. Because $Z_{\lambda}^{\text{OT}}$ now lies within the target decoder's valid density region (Agustsson et al., 2019), the decoded weights remain stable and functional, enabling robust cross-architecture merging even between disparate model families. The overall process for heterogeneous LLMs merging is summarized in algorithm 2. We extend latent-space linear merging from pairs to $N$ models (e.g., LoRA experts) via a convex barycenter. For each expert $W_i$, we sample latent codes $z_i^{(m)} \sim q(z \mid W_i)$. We then merge experts by barycentric interpolation, $z_{\text{merged}} = \sum_{i=1}^{N} \lambda_i z_i$, $\quad \lambda_i \geq 0, \ \sum_{i=1}^{N} \lambda_i = 1$, and decode the unified weights as $\widehat{W}_{\text{merged}} = D(z_{\text{merged}})$. Uniform and Greedy Soup correspond to different choices/updates of $\{\lambda_i\}$.

## 4 EXPERIMENTS

**General Setup** We evaluate latent–space merging on `Gemma-3-1B-it`, `Gemma-3-4B-it`, `Llama-3-1B-instruct`, `Llama-2-7b`, and benchmark latent–space expert fusion against weight–space merging using 10 LoRA experts on `Gemma-7B-it`.

**Datasets and Tasks** We first use subset dataset used by Feng et al. (2024b), such as language understanding (MMLU (Hendrycks et al., 2021), MMLU-pro (Wang et al., 2024b)), commonsense reasoning (HellaSwag (Zellers et al., 2019)), math (GSM8k (Cobbe et al., 2021)), and knowledge-intensive tasks (Knowledge Crosswords (Ding et al., 2024), NLGraph (Wang et al., 2024a), TruthfulQA (Lin et al., 2022), AbstainQA (Feng et al., 2024a)).

Training data consist of pretrained weight snapshots for `Gemma-3-1B-it` and `Gemma-3-4B-it`, plus LoRA experts from Feng et al. (2024b).

**Baselines** We compare against reference-free weight–space methods: spherical linear interpolation (SLERP), uniform soup (Wortsman et al., 2022), greedy soup (Wortsman et al., 2022), data-level

|               | MMLU                | MMLU-pro            | HellaSwag           | Gsm8k               |
| ------------- | ------------------- | ------------------- | ------------------- | ------------------- |
| Gemma-3-4b-it | 53.10               | 20.90               | 47.40               | 29.90               |
| VAE           | $54.10 \pm 0.36$    | $20.80 \pm 0.20$    | $49.03 \pm 0.70$    | $31.27 \pm 0.55$    |
| LS-Merge      | $\mathbf{54.20 \pm 0.00}$ | $\mathbf{21.02 \pm 0.03}$ | $\mathbf{50.10 \pm 0.00}$ | $\mathbf{32.20 \pm 0.05}$ |
| Gemma-3-1b-it | 32.20               | 7.10                | 28.70               | 16.90               |
| VAE           | $32.60 \pm 0.26$    | $7.60 \pm 0.56$     | $28.57 \pm 0.12$    | $16.77 \pm 0.12$    |
| LS-Merge      | $\mathbf{35.13 \pm 0.02}$ | $\mathbf{10.30 \pm 0.20}$ | $\mathbf{31.16 \pm 0.14}$ | $\mathbf{17.50 \pm 0.01}$ |

Table 2: Benchmark scores for pretrained model, VAE, and LS-Merge.

merging, the raw pretrained checkpoints, and the best single expert per task. This excludes approaches that require access to an unmodified base reference model.

**Evaluation Protocols** We evaluate compression and merging in four scenarios:

1. **Self-Merging:** sample multiple codes from one model's latent distribution and merge them.
2. **Expert Merging:** merge LoRA experts in latent space vs. weight space.
3. **Cross-Architecture Merging:** align and merge models with various depths and widths.
4. **Ablation study:** reconstruction or compression behavior and generalization to unseen checkpoints.

**Weight-Encoding Models** We evaluated a Transformer-VAE and a ConvNet-VAE (Soro et al., 2025). We used AdamW (lr $= 1e-4$, weight decay $= 1e-5$) with a cosine-to-zero schedule. The checkpoints are from Hugging Face, and further details are given in the supplement. The learning rate was chosen on the basis of hyperparameter tuning.

## 4.1 SELF-MERGING FOR ENHANCED PERFORMANCE

We investigated a self-merging technique designed to enhance a single model's performance by exploring its learned latent distribution. The process involves encoding a model, sampling multiple latent codes from its posterior distribution, merging these codes into a single representation, and decoding it back into the parameter space. For this experiment, we used a single Transformer-VAE with six encoder and decoder blocks, trained jointly on weights from both `Gemma-3-1B-it` and `Gemma-3-4B-it` with the compression ratio held constant at 2. As shown in Table 2, this approach yields an average performance improvement of $\approx 4\%$ over two key baselines: the original base model and a standard VAE reconstruction from a single latent sample. Notably, the gains are more pronounced on the smaller model, consistent with it having tighter capacity constraints.

## 4.2 MERGING LLM EXPERTS IN LATENT SPACE

Next, we evaluated the primary application of our work: combining specialized LoRA experts. We compared our latent space approach against traditional weight-space interpolation methods using the experts from Feng et al. (2024b). In our method, each expert is encoded, their latent representations are merged, and then the resulting latent vector will be decoded into a single fused model. As shown in Table 3, our latent-space fusion consistently outperforms all weight-space baselines, including both linear and SLERP interpolation across uniform and greedy soup. We found this advantage stems from increased robustness. For example, greedy soup is highly sensitive to initialization, the checkpoint with the best validation accuracy often fails to yield the best test performance. By sampling multiple latent codes for each expert before merging, our method explores the learned parameter distribution instead of relying on a single point estimate, creating more robust and effective combinations.

## 4.3 COMPARISON TO REPRESENTATION-MERGING METHODS

To assess the robustness of our latent space approach, we benchmark it against leading representation-merging methods: Task Arithmetic (Ilharco et al., 2023) and Activation-Informed Merging (AIM) (Nobari et al., 2025), which operate on model activations rather than parameters. contrairily to the previous experiments, in this setting we use *lm-eval* tool (Gao et al., 2024) for fair comparison with the baselines. For this comparison, we merged `Llama-2-13B` models fine-tuned on distinct domains, utilizing a single VAE trained on the combined weights of all constituent models. The results

|  | MMLU | MMLU-pro | HellaSwag | Gsm8k | TruthfulQA | NLGraph | K-Crossword | AbstainQA |
|---|---|---|---|---|---|---|---|---|
| Best expert | 45.7 | 14.3 | 46.6 | 26.1 | 32.4 | 51.7 | 32.7 | -10.8 |
| Base model | 48.8 | 18.1 | 53.3 | 6.9 | 30.1 | 47.5 | 28.0 | -0.9 |
| Data Merge[*] | 44.5 | 17.6 | 52.7 | 14.3 | 10.7 | 42.3 | **37.0** | -2.5 |
| Uniform Soup | 49.7 | 19.4 | 54.0 | 7.9 | 31.2 | 47.5 | 29.6 | -0.1 |
| SLERP(t=0.45) | 52.5 | 18.8 | 50.4 | 25.5 | 28.7 | 49.8 | 30.0 | -0.2 |
| Greedy Soup | 50.8 | 22.1 | 54.6 | 23.9 | 31.9 | 52.9 | 28.0 | 3.3 |
| Dare-Ties | 49.1 | 18.8 | 53.7 | 7.3 | 28.2 | 52.8 | 29.0 | 1.4 |
| LS-Merge(lerp) | 54.7 | 21.6 | 58.1 | **28.1** | **33.0** | 53.1 | 35.6 | 2.0 |
| LS-Merge(soup) | **56.0** | **22.2** | **60.1** | 24.2 | 32.0 | **56.1** | 35.2 | **4.0** |

Table 3: Results on merging expert LoRA weights in the raw weight space and the latent space.

| Method | MMLU | IFEval | MBPP | HumanEval | GSM8k |
|---|---|---|---|---|---|
| base | 52.18 | 25.10 | 27.80 | 17.07 | 4.20 |
| code | 52.91 | 26.25 | 31.60 | 17.07 | 24.10 |
| instruct | 53.41 | 35.67 | 34.80 | 26.83 | 43.40 |
| code + instruct (Task Arithmetic) | 52.18 | 25.10 | 34.40 | 26.83 | 4.20 |
| code + instruct (AIM) | 54.18 | 32.00 | 36.00 | **29.27** | **46.20** |
| code + instruct (LS-merge) | **55.07** | **36.41** | **36.02** | 28.14 | 44.12 |

Table 4: Comparison of our latent space merging (LS-merge) with Task Arithmetic and AIM on `Llama-2-13B` fine-tuned models.

presented in Table 4 demonstrate that our method is highly competitive. It achieves performance comparable to the state-of-the-art AIM and substantially outperforms Task Arithmetic. This finding is significant, as it shows that a latent weight-space approach can match the performance of prominent methods that require access to model activations.

## 4.4 Cross-Architecture Merging

Our latent-space approach supports merging across substantial architectural gaps. We study two settings: (i) *intra-family* (different sizes within Gemma) and (ii) *cross-family* (Gemma ↔ LLaMA).

**Intra-family (Gemma-3-4B-it → Gemma-3-1B-it).** Direct latent interpolation between independently trained models is unstable (Fig. 4b). Aligning the larger model's latents to the smaller model's distribution before interpolation yields consistent gains across the mixing range. As shown in Fig. 4a, small injections from the source ($\lambda \in [0.05, 0.20]$) deliver the best improvements over the `Gemma-3-1B-it` baseline.

**Cross-family (LLaMA-3.2-1B-instruct → Gemma-3-1B-it).** Family-level transfer is more challenging: baseline parameter/latent mixing without alignment degrades performance. Table 5 shows that distributionally aligned latent merging recovers and surpasses these baselines, a modest interpolation weight ($\lambda = 0.1$) achieves the best overall scores. The evaluation for cross family evaluation is performed using *lm-eval* for simplicity and also due to some issues with llama model when using the previous evaluation code.

**Takeaway.** *Matching latent dimensionality* is insufficient for heterogeneous merges, *aligning latent distributions* is crucial. Once aligned, a single knob $\lambda$ reliably controls how much capacity is injected from the source into the target.

## 5 Ablation Studies

To better understand the properties of our method, we conduct two targeted ablation studies. In this section we use *lm-eval* for evaluation unless stated.

## 5.1 Component Contributions in Latent Merging

We first analyze how different submodules contribute to the merged model's performance by merging MLP layers only, self-attention layers only, or both jointly. The results in Table 6 show that merging MLP layers alone provides modest gains, while merging attention layers alone degrades performance. Optimal results are achieved by merging both, indicating that **MLP and self-attention parameters**

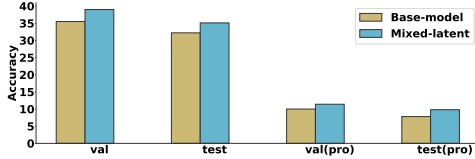

(a) Heterogeneous latent space merging.

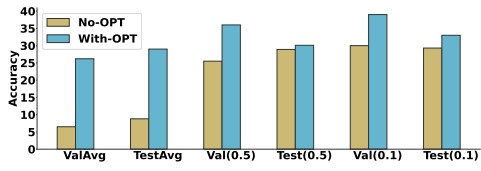

(b) Varying coefficient on MMLU.

Figure 4: (a) Accuracy gains obtained by latent space merging of `Gemma-3-4B-it` with `Gemma-3-1B-it`. Bars show performance on the validation (*val*) and test splits of the MMLU and MMLU-PRO benchmarks (b) Impact of the merging coefficient $t$ on validation and test accuracy in latent-space merging.

| Strategy | WinoGrande | ARC-C | HellaSwag |
|---|---|---|---|
| Base | 56.83 | 42.78 | 49.07 |
| OT only | 51.13 | 34.25 | 48.50 |
| OT + interp. | **57.75** | **43.34** | **50.10** |

Table 5: Downstream accuracy on WinoGrande, ARC-Challenge, and HellaSwag for different alignment strategies with $\lambda = 0.1$.

| Strategy | WinoGrande | ARC-C | MMLU |
|---|---|---|---|
| Base | 56.83 | 42.78 | 40.76 |
| MLP | 56.84 | **43.89** | 41.02 |
| Attention | 56.67 | 40.23 | 39.80 |
| Attention + MLP | **57.75** | 43.34 | **42.10** |

Table 6: Ablation: Merging subsets of layers. MLP-only outperforms on ARC-C, combining MLP and Attention achieves the best results.

**encode complementary functional knowledge**, and that altering one without the other can disrupt learned co-adaptations.

## 5.2 VAE GENERALIZATION AND THE COMPRESSION TRADE-OFF

Next, we assess the VAE's zero-shot generalization by training it on `Gemma-3-4B-it` and evaluating it on two unseen models: an in-family `Gemma-3-1B-it` and an out-of-family `LLaMA-3.2-1B-it`. Table 7 reveals a clear trade-off between compression and generalization. At a low compression ratio ($r = 1.6$), the VAE maintains strong performance on both unseen models. However, performance degrades substantially at higher ratios ($r = 2, 4$). This suggests that while the VAE learns transferable representations of weight structures, higher compression ratios lead to posterior collapse due to the fact most of the data sample are cluster aroung zero.

## 5.3 LINEAR SUBSPACE VS. NON-LINEAR MANIFOLD LEARNING

In this section we investigate the use of incremental PCA for weights encoding on gemma-3-1 b-it compare to VAE based en coding. Although Section 3.1 showed that LLM weight matrices exhibit low-rank structure, this does not imply that the space of functional parameters forms a linear subspace. To assess whether linear methods are sufficient, we compare our non-linear Transformer-VAE against PCA across compression ratios $r \in \{1.6, 2.0, 4.0\}$ (Table 8).

**PCA collapses functional performance.** Across all ratios, PCA-reconstructed models regress to near-random accuracy on MMLU ($\approx 25.5\%$ at $r = 1.6$) and exhibit a global drop across benchmarks (e.g., ARC-C: $42.41\% \rightarrow 27.65\%$). Notably, performance is equally poor at $r = 1.6$ and $r = 4.0$, indicating that the failure is not due to insufficient latent capacity but to a structural mismatch: the set of valid pretrained weights does not reside in a linear subspace.

**VAE preserves the functional manifold.** In contrast, the LS-Merge VAE retains near-original accuracy at all compression levels. At $r = 1.6$, it recovers 96% of the base model's MMLU performance (39.89% vs. 41.44%) and even improves Winogrande (56.64% vs. 55.41%). Remarkably, VAE reconstructions remain stable at $r = 4.0$, whereas PCA has already collapsed at $r = 1.6$. This indicates that pretrained weights lie on a non-linear manifold that requires expressive encoders and decoders to model its curvature. Linear projections such as PCA cannot preserve the structure of the pretrained weight manifold and produce functionally invalid models even under mild compression. The VAE's non-linear latent geometry is therefore not a stylistic preference but a geometric necessity for faithful reconstruction, compression, and downstream operations such as interpolation and merging.

| Model | Winogrande (5-shot) | ARC-Challenge (25-shot) | HellaSwag (10-shot) | MMLU (5-shot) |
|---|---|---|---|---|
| Gemma-3-1B-it (base) | 56.83 | 42.78 | 49.07 | 40.76 |
| LLaMA-3.2-1B-it (base) | 61.56 | 41.11 | 61.62 | 46.55 |
| Gemma-3-1B-it (VAE, $r = 1.6$) | 56.67 | 42.83 | 47.31 | 39.98 |
| LLaMA-3.2-1B-it (VAE, $r = 1.6$) | 61.25 | 41.47 | 60.80 | 46.06 |
| Gemma-3-1B-it (VAE, $r = 2$) | 56.67 | 38.23 | 38.88 | 32.22 |
| LLaMA-3.2-1B-it (VAE, $r = 2$) | 59.43 | 37.12 | 55.99 | 39.73 |
| Gemma-3-1B-it (VAE, $r = 4$) | 46.49 | 28.24 | 25.66 | 25.02 |
| LLaMA-3.2-1B-it (VAE, $r = 4$) | 49.20 | 26.28 | 25.70 | 26.76 |

Table 7: Accuracy of VAE models (trained on `Gemma-3-4B-it`) when evaluated on `Gemma-3-1B-it` and `LLaMA-3.2-1B-it`. Performance remains stable for $r = 1.6$, but deteriorates as the compression factor increases.

Table 8: Functional Reconstruction Fidelity vs. Compression Ratio. Zero-shot accuracy comparison on standard benchmarks (MMLU, HellaSwag, Winogrande, ARC-C) for Gemma-3-1B-it. We compare our non-linear Transformer-VAE (LS-Merge) against a linear PCA baseline at compression ratios $r \in \{1.6, 2, 4\}$. While the VAE maintains strong performance at $r = 1.6$ and $r = 2$, linear compression suffers significantly as the bottleneck tightens, validating the need for non-linear manifold learning.

| Method | Ratio ($r$) | MMLU | HellaSwag | Winogrande | ARC-C |
|---|---|---|---|---|---|
| Gemma-3–1b-it) | 1.0× | $41.44 \pm 0.00$ | $49.05 \pm 0.01$ | $55.41 \pm 0.03$ | $42.41 \pm 0.02$ |
| PCA (Linear) | 1.6× | $25.50 \pm 0.37$ | $25.56 \pm 0.04$ | $50.12 \pm 0.01$ | $27.65 \pm 0.01$ |
| LS-Merge VAE | 1.6× | $\mathbf{39.89} \pm 0.01$ | $\mathbf{48.57} \pm 0.25$ | $\mathbf{56.64} \pm 0.15$ | $\mathbf{41.64} \pm 0.01$ |
| PCA (Linear) | 2.0× | $24.12 \pm 0.00$ | $25.27 \pm 0.12$ | $46.27 \pm 0.01$ | $26.24 \pm 0.32$ |
| LS-Merge VAE | 2.0× | $\mathbf{39.80} \pm 0.00$ | $\mathbf{49.29} \pm 0.10$ | $\mathbf{54.14} \pm 1.02$ | $\mathbf{42.32} \pm 0.21$ |
| PCA (Linear) | 4.0× | $24.13 \pm 0.15$ | $24.79 \pm 0.23$ | $49.57 \pm 0.01$ | $25.89 \pm 0.23$ |
| LS-Merge VAE | 4.0× | $\mathbf{39.83} \pm 0.00$ | $\mathbf{49.30} \pm 0.21$ | $\mathbf{56.06} \pm 0.15$ | $\mathbf{42.75} \pm 0.20$ |

## 6 DISCUSSION

**Limitations** Despite its strong performance, training the weight-encoding VAE at high compression ratios can be challenging(mode collapse), largely due to the heavy-tailed nature of LLM weight distributions. Nonetheless, our merging paradigm does not strictly require a tight bottleneck; it remains highly effective when utilizing an overcomplete latent space, which eases optimization while successfully unfolding the weight manifold.

This work opens several avenues for future research. Promising directions include developing more efficient, property-aware weight encoders (Navon et al., 2023; Zhou et al., 2023) to improve scalability. Additionally, extending our framework beyond simple merging via iterative latent barycentric interpolation (Akash et al., 2022; Pennec, 2017) could unlock complex uni-model compositions, paving the way for advanced multimodal generation.

## 7 CONCLUSION

In this work, we introduced LS-Merge, a novel framework that reimagines model merging by operating in a learned latent space of model parameters. By mapping weights to a continuous manifold and critically employing OT for principled alignment, our method successfully merges models with heterogeneous architectures, overcoming a fundamental limitation of prior weight-space techniques. Our comprehensive experiments demonstrate that this approach not only excels at standard expert fusion but also enables robust cross-scale and cross-family model merging for the first time. LS-Merge establishes a scalable and architecture-agnostic paradigm for model composition, paving the way for more flexible and efficient reuse of pre-trained models.

## REPRODUCIBILITY STATEMENT

To ensure reliable and reproducible results, we have provided detailed experiment settings in the Appendix. We plan to open-source our implementation here: https://github.com/sorobedio/ls-merge/.

## ACKNOWLEDGMENTS

This work was supported by the Institute for Information & Communications Technology Planning & Evaluation (IITP) grant funded by the Korea government (MSIT) (RS-2019-II190075, Artificial Intelligence Graduate School Program (KAIST)); by the National Research Foundation of Korea (NRF) grant funded by the Korea government (MSIT) (No. RS-2023-00256259); and by the Institute of Information & Communications Technology Planning & Evaluation (IITP) grant funded by the Korea government (MSIT) (No. RS-2022-II220713, Meta-learning Applicable to Real-world Problems). This research was also supported by a grant from the Korea Machine Learning Ledger Orchestration for Drug Discovery Project (K-MELLODDY), funded by the Ministry of Health & Welfare and the Ministry of Science and ICT, Republic of Korea (Grant No. RS-2024-00460870); and by the Institute of Information & Communications Technology Planning & Evaluation (IITP) under the Open RAN Education and Training Program (IITP-2024-RS-2024-00429088), funded by the Korea government (MSIT).

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

# A APPENDIX

## IMPACT STATEMENT

We introduce a VAE-based latent-space merging technique for pretrained LLMs that encodes weights into compact codes, blends them, and decoding new high-performance parameters. This enables fast, compute-efficient model customization and smooth interpolation across architectures. However, blending latent codes can unintentionally merge biases or toxic behaviors, obscure the origin of capabilities, and be misused to graft malicious functionality. We advocate for rigorous bias / toxicity audits, transparent provenance tracking, and clear reporting guidelines to ensure responsible deployment.

# A EXTENDED DISTRIBUTION ANALYSIS

Section 3.1 characterized layerwise weight statistics to inform encoding and merging. Here we extend that analysis with (i) complete per-layer distribution plots, including all *MLP* layers for GEMMA-3-1B-IT, GEMMA-3-4B-IT, and LLAMA-3.2-3B-INSTRUCT, and (ii) cumulative variance–explained curves from PCA on those MLP layers.

## A.1 DATA AND PROCEDURE

**Models.** GEMMA-3-1B-IT, GEMMA-3-4B-IT, LLAMA-3.2-3B-INSTRUCT, plus the main-paper models.

**Layer selection.** For each transformer block, we analyze *self-attention* (Q/K/V/O projections) and *MLP* (up/gate/down) weight matrices independently.

**Preprocessing.** Unless noted, statistics are computed on raw matrix entries. For PCA, matrices are mean-centered and the spectrum is computed over flattened rows; we report cumulative variance explained. We use *excess kurtosis* (Fisher convention) so a Gaussian has $\kappa_{\text{ex}}=0$:

$$\kappa_{\text{ex}} = \frac{\mathbb{E}[(W - \mu)^4]}{\sigma^4} - 3.$$

Means and variances are standard sample estimates; skewness uses the unbiased estimator.

**Aggregation.** We report per-layer curves and layer-type aggregates (self-attention vs. MLP). Family summaries first average across layers within a model, then across models in the same family.

## A.2 COMPLETE PLOTS

Figure 5 shows layerwise distribution plots (self-attention and MLP). Figure 6 reports cumulative PCA variance for all MLP layers (attention spectra are in the main text).

## A.3 FINDINGS AND IMPLICATIONS

**Self-attention layers exhibit heavier tails in Gemma.** In self-attention projections (Fig. 7a, Table 9), GEMMA-3-1B-IT and GEMMA-3-4B-IT show pronounced positive excess kurtosis with depth-localized peaks, indicating heavy tails and more extreme outliers. LLAMA models track closely across scales, suggesting family-level statistical stability.

**MLP layers are more stable and closer to Gaussian.** Across models (Fig. 7b, Table 9), MLP projections are near-Gaussian with excess kurtosis typically in $[0, 2]$ and fewer depth-dependent spikes. Their PCA spectra (Fig. 6) decay faster than attention, indicating lower intrinsic dimensionality.

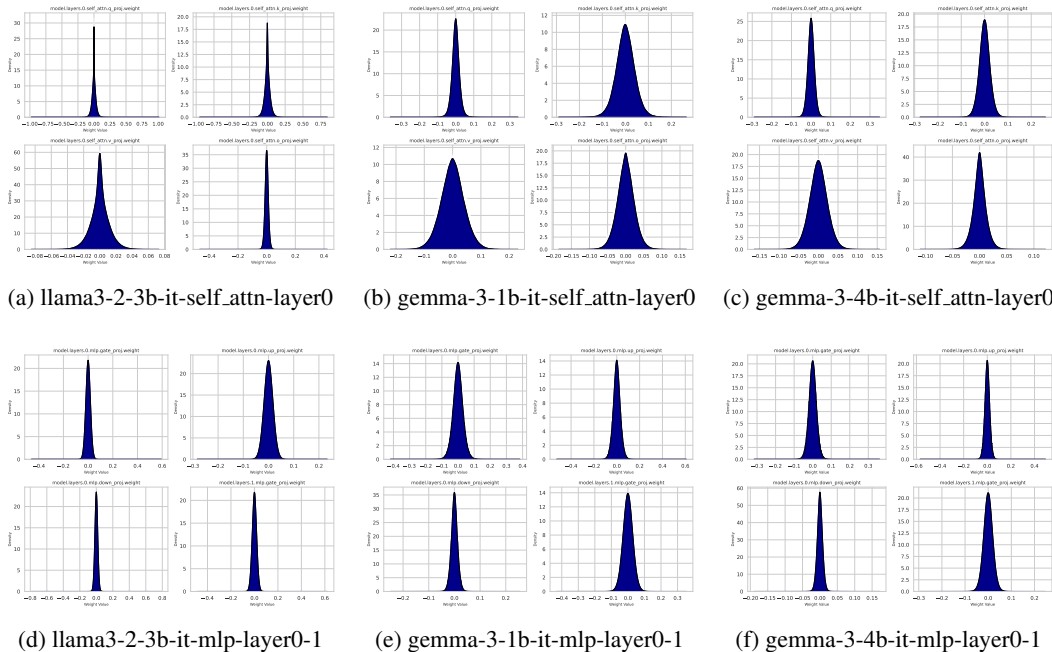

(a) llama3-2-3b-it-self_attn-layer0    (b) gemma-3-1b-it-self_attn-layer0    (c) gemma-3-4b-it-self_attn-layer0

(d) llama3-2-3b-it-mlp-layer0-1    (e) gemma-3-1b-it-mlp-layer0-1    (f) gemma-3-4b-it-mlp-layer0-1

Figure 5: Distribution plot of the self-attention and the mlp modules in llama3-1-8b-instruct, gemma-3-1b-it, and gemma-3-4b-it for attention layer 0.

| layers | llama3-2-3b-it | | | gemma-3-1b-it | | | gemma-3-4b-it | | |
|---|---|---|---|---|---|---|---|---|---|
| | var. | skew | kurt. | var. | skew | kurt. | var. | skew | kurt. |
| self_attn | 0.0017 | 0.0192 | 8.4032 | 0.0031 | 0.0500 | 15.0505 | 0.0015 | 0.0501 | 15.2010 |
| | 0.0013 | 0.0191 | 7.3438 | 0.0030 | 0.0450 | 9.8347 | 0.0015 | 0.0267 | 7.3832 |
| | 0.0012 | 0.0178 | 6.2172 | 0.0030 | 0.0389 | 9.0363 | 0.0014 | 0.0255 | 7.2277 |
| | 0.0011 | 0.0104 | 5.4477 | 0.0030 | 0.0288 | 8.6496 | 0.0014 | 0.0172 | 6.0731 |
| avg (self_attn) | 0.0005 | -0.0002 | 1.4342 | 0.0012 | -0.0009 | 3.2858 | 0.0005 | 0.0009 | 2.6900 |
| min (self_attn) | 0.0001 | -0.0131 | 0.3027 | 0.0002 | -0.0418 | 0.1589 | 0.0001 | -0.0305 | 0.2412 |
| mlp | 0.0005 | 0.0093 | 5.4740 | 0.0010 | 0.0167 | 8.7665 | 0.0004 | 0.0266 | 6.3297 |
| | 0.0005 | 0.0080 | 4.1465 | 0.0010 | 0.0090 | 3.1514 | 0.0004 | 0.0155 | 5.9670 |
| | 0.0004 | 0.0075 | 2.6694 | 0.0010 | 0.0077 | 3.0577 | 0.0004 | 0.0130 | 2.7454 |
| | 0.0004 | 0.0074 | 2.4364 | 0.0010 | 0.0071 | 3.0113 | 0.0004 | 0.0094 | 2.4371 |
| avg (mlp) | 0.0003 | 0.0003 | 0.8435 | 0.0006 | 0 | 1.1739 | 0.0003 | 0.0006 | 1.0807 |
| min (mlp) | 0.0003 | -0.0139 | 0.0892 | -0.0001 | -0.0184 | 0.1552 | 0 | -0.0076 | 0.1559 |

Table 9: Statistical moments of the self-attention and MLP layers across three models.

**Design implications for encoding and merging.**

- *Allocate capacity to attention.* Heavy tails in attention (notably in Gemma) warrant encoders with higher capacity or robust priors; Gaussian assumptions under-represent extremes.

- *MLP is the easy regime.* More Gaussian, stable MLP statistics admit accurate compression with standard VAE settings and fewer latent pathologies.

- *Family consistency aids calibration.* The alignment of LLAMA statistics across scales simplifies cross-scale latent calibration and reduces merging friction.

**Summary.** Attention weights (especially in Gemma) are the dominant source of heavy-tail behavior; MLP weights are comparatively benign. Tail-aware encoding and depthwise calibration are most critical for attention, while default settings suffice for MLP.

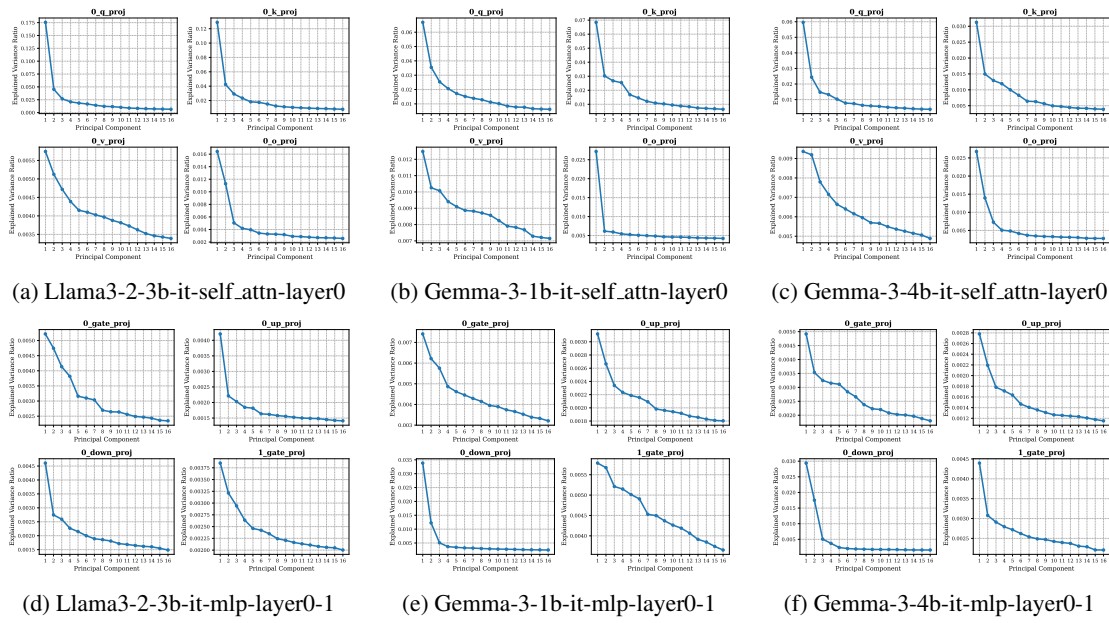

(a) Llama3-2-3b-it-self_attn-layer0

(b) Gemma-3-1b-it-self_attn-layer0

(c) Gemma-3-4b-it-self_attn-layer0

(d) Llama3-2-3b-it-mlp-layer0-1

(e) Gemma-3-1b-it-mlp-layer0-1

(f) Gemma-3-4b-it-mlp-layer0-1

Figure 6: Plots of the PCA **explained-variance ratio** for individual weight matrices in the first self-attention block (top row of each subpanel set) of three LLM checkpoints—Llama-3-2-3b-it, Gemma-3-1b-it, and Gemma-3-4b-it. For each model, we show the four projection matrices of self-attention (*q, k, v, o*). The sharp drop after the leading principal components highlights a pronounced low-rank structure that is consistent across architectures and model sizes.

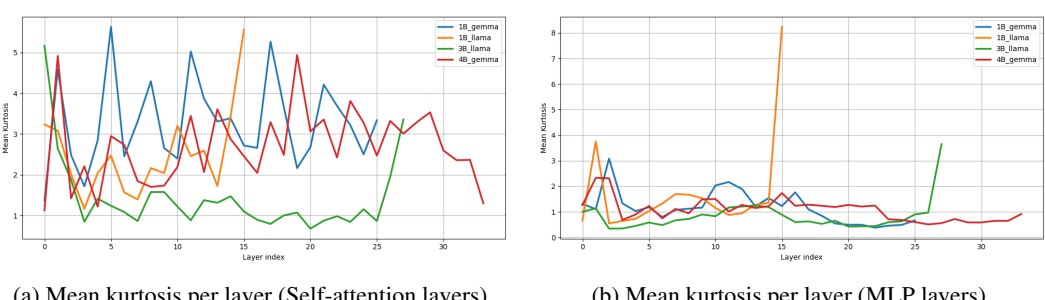

(a) Mean kurtosis per layer (Self-attention layers).

(b) Mean kurtosis per layer (MLP layers).

Figure 7: Comparison of mean kurtosis across layers in self-attention and MLP components for Gemma and LLaMA models.

## B  VARIATIONAL ENCODER ARCHITECTURE AND TRAINING DETAILS

Table 10 lists the configuration of our transformer VAE. The encoder applies a twofold down-projection of the token (*compression ratio r=2*), which is the smallest bottleneck we found that preserves reconstruction while generalizing to unseen checkpoints. The decoder mirrors the encoder to restore the original dimension. For heterogeneous merges, we set the latent size $d_z$ by the source to target layer mapping ratio and keep all other hyperparameters fixed, ensuring comparable latent scales across architectures.

**Objective.** We optimize the ELBO (reconstruction + KL to $\mathcal{N}(0, I)$) with a constant learning rate $1\times10^{-5}$ for 10,000 epochs.

**Training setup.** Single NVIDIA A6000, bfloat16 for forward and backward, the remaining settings (optimizer, batch size, weight decay, clip-norm) are in Table 10. Seeds are fixed for data order and initialization.

| Field | Value | Notes |
|---|---|---|
| `length` | 10,240 | Sequence length processed |
| `n_layers` | 6 | Transformer depth |
| `chunk_size` | 640 | Per-token chunk width |
| `embed_dim` | 768 | Matches global `embed_dim` |
| `latent_dim` | 640 | Size of latent vector per chunk |
| `n_heads` | 4 | Attention heads |
| `rope_base` | 20,000 | RoPE base frequency |
| `conv` | false | No convolutional patching |
| `flatten` | true | Flatten to original shape. |

Table 10: Transformer block settings shared by the encoder and decoder.

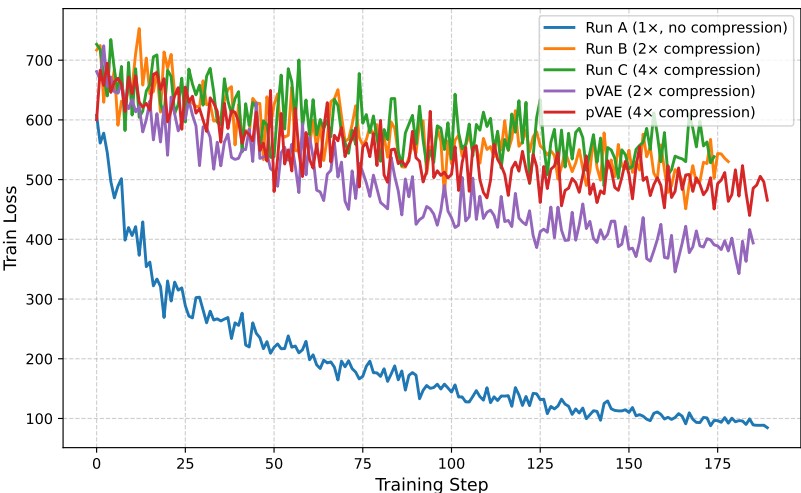

Figure 8: Mean reconstruction error curves (smoothed with a 5-step moving average) for pVAE (VAE initialized from an existing pretrained VAE with reduction factor 1) and VAE (random initialized VAE). $r$ denotes the compression ratio.

**Ablation summary.** Bottlenecks weaker than $r{=}2$ overfit and degrade out-of-distribution reconstruction, stronger bottlenecks raise reconstruction error and harm downstream merging. Choosing $d_z$ via the mapping ratio stabilizes cross-architecture alignment without retuning.

### B.1 ANALYSIS WEIGHTS ENCODING EVOLUTION

Pretrained weight matrices exhibit substantial variation in tail behavior (high to near-zero excess kurtosis; cf. Section 3.1). We study how this variability interacts with the compression ratio $r$ in the VAE and how it impacts generalization to unseen checkpoints.

**Convergence vs. compression.** Figure 8 shows a clear monotone effect: optimization slows as the bottleneck tightens. With $r{=}2$ (twofold down-projection), storage is halved and the reconstruction loss increases by only $\approx 6\%$; at $r{=}4$, the penalty rises to $\approx 10\%$. These curves set a practical operating point: $r{\approx}2$ balances footprint and fidelity with minimal training overhead.

**Sensitivity to distribution shape.** Layers with heavier tails (high excess kurtosis) are more sensitive to increasing $r$, exhibiting larger loss gaps at fixed training budget, near-Gaussian layers degrade more gently. In our models, attention projections in GEMMA fall into the former regime, whereas MLP projections across families align with the latter (App. A).

**Training schedule.** Two-Stage Training. To improve stability under tight bottlenecks, we employ a two-stage curriculum: (1) train a deterministic autoencoder to convergence with the KL term disabled, then (2) enable KL regularization and fine-tune as a VAE. This approach prevents early posterior

---

**Algorithm 2** Details of Heterogeneous LLM Parameter Merging in Latent Space

---

**Require:** Source weights $W_{\text{src}}$, target weights $W_{\text{tgt}}$, VAE configs $C_{\text{src}}, C_{\text{tgt}}$
**Ensure:** Merged weights $W_{\text{merged}}$
 1: Initialize VAEs: $(V_{\text{src}}, V_{\text{tgt}}) \leftarrow init(C_{\text{src}}, C_{\text{tgt}})$
 2: Load pretrained parameters into $V_{\text{src}}, V_{\text{tgt}}$
 3: Split weights into per-layer groups:
   $L_{\text{src}} \leftarrow \{\text{all layers from } W_{\text{src}}\}$
   $L_{\text{tgt}} \leftarrow \{\text{all layers from } W_{\text{tgt}}\}$
 4: Determine the number of pairs to merge: $N \leftarrow \min(|L_{\text{src}}|, |L_{\text{tgt}}|)$
 5: Define aligned layer pairs: $(l_{\text{src}}^{(j)}, l_{\text{tgt}}^{(j)})$ for $j = 1, \dots, N$
 6: **for** each aligned pair $(l_{\text{src}}^{(j)}, l_{\text{tgt}}^{(j)})$ **do**
 7:    Flatten & chunk weights into $\{w_{\text{src}}^{(i)}, w_{\text{tgt}}^{(i)}\}$
 8:    Encode: $z_{\text{src}}^{(i)} \leftarrow V_{\text{src}}(w_{\text{src}}^{(i)}), z_{\text{tgt}}^{(i)} \leftarrow V_{\text{tgt}}(w_{\text{tgt}}^{(i)})$
 9:    Align latents via OT: $z_{\text{align}}^{(i)} \leftarrow OT(z_{\text{src}}^{(i)}, z_{\text{tgt}}^{(i)})$
10:    Merge latents: $z_{\text{merged}}^{(i)} \leftarrow z_{\text{tgt}}^{(i)} + \beta \cdot (z_{\text{align}}^{(i)} - z_{\text{tgt}}^{(i)})$
11:    Decode: $w_{\text{merged}}^{(i)} \leftarrow V_{\text{tgt}}^{-1}(z_{\text{merged}}^{(i)})$
12:    Store $w_{\text{merged}}^{(i)}$ in $W_{\text{merged}}$
13: **end for**
14: Initialize and evaluate final network with complete $W_{\text{merged}}$
15: **return** $W_{\text{merged}}$

---

collapse and yields stable convergence even at high compression ratios, without increasing total training time.

**Computational Cost**   All experiments were conducted on a single NVIDIA RTX 6000 Ada GPU. Training a $\sim 200M$ parameter VAE at compression ratio $r = 1.6$ requires approximately 1-2 hours for 1B-scale models and 3–4 hours for 7B-scale models ($\sim$500K chunks). Higher compression ratios increase training time proportionally. LoRA weights converge notably faster than full model weights due to their lower-rank structure. At inference, the full encode-decode pipeline completes in 5–10 seconds, enabling rapid iteration during merging experiments.

**Reducing the Training Time:**    Interestingly, in the case of heterogeneous models expansion $r < 1$ can offer an alternative trade-off. Lifting weights into an overcomplete latent space can unfold curved manifolds, making linear interpolation better approximate geodesics on the original weight manifold. expansion forces the encoder to learn a non-trivial transformation, avoiding identity collapse. Practically, expansion significantly accelerates training the relaxed bottleneck eases optimization and improves gradient flow at the cost of increased memory.

## C  HETEROGENEOUS MODEL MERGING IN LATENT SPACE

Merging models with matched layer shapes is straightforward in weight space. Heterogeneous pairs are harder: to avoid truncating the higher-capacity model, we train *separate* VAEs per architecture and merge in a shared, *aligned* latent space. Concretely, we first calibrate latents per layer, fit an alignment map between the source and target latent spaces, and then interpolate on the target side before decoding. This preserves each model's capacity while enabling stable cross-architecture merges without ad hoc dimensionality cuts. The full procedure is given in Algorithm 2.

### C.1  INTUITION BEHIND LATENT SPACE ALIGNMENT

The rationale for restricting explicit latent alignment specifically to heterogeneous merging scenarios relies on the geometric properties of the neural loss landscape and the topological structure of the parameter manifolds.

**Homogeneous Merging and Linear Mode Connectivity.** In the case of homogeneous models—specifically those finetuned from a shared pretrained initialization (e.g., distinct fine-tunes of LLama-2-13B)—the parameters remain within a shared basin of attraction. Consequently, their weight vectors $\theta_1, \theta_2$ reside on a connected region of the high-dimensional parameter manifold $\mathcal{M}$. When encoded into the latent space $\mathcal{Z}$, the resulting distributions $p(z|\theta_1)$ and $p(z|\theta_2)$ naturally share a common support and overlapping density. Therefore, interpolation in $\mathcal{Z}$ corresponds to traversing a flat, low-loss path on the underlying manifold, rendering explicit distributional alignment redundant.

**The Heterogeneous Disconnect.** Conversely, heterogeneous models (e.g., Gemma-3-1B vs. LLaMA-3.2-1B) possess fundamentally distinct architectures. Even when projected into a latent space of identical dimensionality $d$, their representations occupy disjoint manifolds $\mathcal{M}_{src}$ and $\mathcal{M}_{tgt}$ with divergent geometric structures and density profiles as show in Figure 9

This disjoint nature necessitates a non-linear mapping to bridge $\mathcal{M}_{src}$ and $\mathcal{M}_{tgt}$. We employ Optimal Transport (specifically the Monge map) to push the source distribution $\mu_{src}$ onto the target distribution $\mu_{tgt}$ by minimizing the Wasserstein-2 distance. This process essentially "registers" the two manifolds, aligning their statistical moments and geometric structure. By enforcing this alignment, we ensure that the interpolated latent codes $z_\alpha$ remain within the valid density region of the target decoder, thereby preserving functional competence across architectural boundaries.

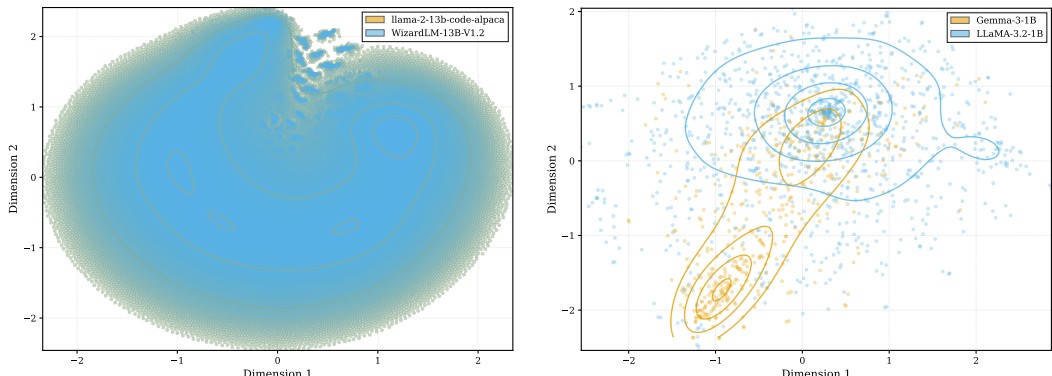

(a) Same-Architecture Fine-tuning Variants: LLaMA-2-13B Code-Alpaca vs WizardLM-13B-V1.2.

(b) Cross-Architecture Comparison: Gemma-3-1B vs LLaMA-3.2-1B.

Figure 9: **VAE latent space reveals architectural signatures in weight distributions.** t-SNE projections of encoded k_proj weights show (**a**) fine-tuned variants of the same base model (LLaMA-2-13B) remain indistinguishable while (**b**) distinct clusters for different architectures (Gemma vs LLaMA) , suggesting the learned latent space captures intrinsic architectural properties.

