# OpenReview forum: "LS-Merge: Merging Language Models in Latent Space"
_ICLR.cc/2026/Conference — ICLR 2026 Poster_

### Official Review · Reviewer_mkDJ · 2025-10-30

**Soundness:** 3
**Presentation:** 3
**Contribution:** 4
**Rating:** 8
**Confidence:** 3

**Summary:**

This paper propose a novel method to replace fragile weight-space merging with LSMerge: a latent space merging technique. It is an encode–align–decode pipeline that operates on a latent space of model weights learned by a transformer VAE. Specifically: the method
(1) encodes parameters into latents via layer-aware chunking,
(2) aligns heterogeneous models’ latent distributions (depth/width mismatches) using Optimal Transport (OT), and
(3) decodes an interpolated latent back to weights.
Evaluations demonstrates performance gain in (i) self-merging of a single model’s latents, (ii) LoRA expert fusion vs. model-soup/SLERP baselines, and (iii) heterogeneous merges (within-family size changes and cross-family).

**Strengths:**

1. Principled formulation: The core idea of encoding model weight into a shared latent manifold and aligning distributions via OT is right on target to resolve the pain point of current merging methods in shape matching requirements. This paper gives a more grounded understanding of how to merge models by aligning the model's representation geometry.

2. Comprehensive experiment setting: The same recipe can be used for many use cases, including self-merging, LoRA-expert fusion, and cross-family merging. This shows that this method is a more fundamental framework rather than a "trick".

**Weaknesses:**

1. Lack of analysis on reconstruction error: The main component of the merging system is a VAE which introduces lossy compression during reconstruction, yet there isn't an analysis of how that bottleneck correlates with downstream task accuracy. This matters as it could confound where the gains are from. I'm curious if the author can ablate on compression level vs. KL-weights.

2. Lack of understanding of successful merging condition: The experiments are diverse in terms of merging various types of models (self, LoRA etc.) but it is unclear what is the boundary condition of successful latent space merging? I'm curious to see if the authors have found any failure case and if so, have done any analysis of what's the key difference between merges that fails/succeeds.

**Questions:**

See weaknesses.

---

> ### Author Response · Authors · 2025-11-19
>
> We sincerly appreciate you for your time and constructive comments which improve our paper. We respectfully address your concerns as following:
>
> ---
>
> **W1** **Reconstruction error and VAE bottleneck**
>
> 1. **Compression ratio (r) vs. downstream accuracy.**
>    In Section 5.2 and the supplementary material (Table 7), we explicitly vary the compression ratio (r) and evaluate performance on *unseen* checkpoints. These experiments show that larger (r) systematically worsens reconstruction quality and leads to clear degradation on downstream tasks, directly linking the VAE bottleneck to task accuracy.
>
> 2. **Where the gains come from.**
>    Tables 2 and 3 compare the original model, its VAE sampling, and our merged model. The merged models consistently outperform the VAE-only, indicating that the improvements do not come from sampling alone. A reasonably accurate, generalizable VAE is necessary for good merging performance and a poor VAE may yield to poor merges. We believe that the additional gains are due to the latent-space merging itself with generalizable VAE.
>
> 3. **KL-weight vs. compression.**
>    In our experiments, larger KL-weights (for fixed (r)) lead to slow convergence, increase training stochasticity, and generally worsen reconstruction quality; they do not compensate for the loss of information caused by stronger compression. In the revised version, we will add a small ablation over KL-weights at fixed (r) to illustrate this effect.
>
> **W2**  **Conditions for successful latent-space merging**
>
> 1. When LS-Merge works
>
>    * The VAE, trained on pretrained checkpoints, generalizes to other pretrained models with only a small drop in downstream accuracy.
>    * In this regime, the decoder can reliably map merged latent codes back to meaningful weight configurations.
>
> 2. Boundary conditions for success
>
>    * We require that VAE reconstructions of both training and unseen models stay close to the base model’s performance.
>    * When this holds, LS-Merge typically matches or improves standard weight-space merging methods.
>
> 3. Failure cases and why
>
>    * Failures occur when the VAE cannot reconstruct unseen models well; merged models then struggle to outperform the base model or other merging baselines.
>    * Empirically, this breakdown appears quickly as the compression ratio increases, which harms VAE generalization and leads to distorted merges.

---

### Official Review · Reviewer_593M · 2025-11-01

**Soundness:** 3
**Presentation:** 1
**Contribution:** 3
**Rating:** 4
**Confidence:** 3

**Summary:**

LS-Merge is a method that uses a Transformer-based VAE to compress chunks of weight matrices from various LLMs, enabling heterogenous merges where models have different architectures. THhe VAE encodes weight chunks into lower dimensional representations, aligns them in latent space, interpolates them, and then decodes them back into model space. This method can improve performance of single models, improve merges of multiple LoRA adapters, and perform competitively with direct weight space merging methods.

**Strengths:**

1. By applying the LS-Merge VAE on pre-trained Gemma models, the output model can improve on benchmarks like GSM8K and MMLU without additional gradient steps on the model weights, just on encoding and decoding the weights with the pretrained VAE. This represents a unique and nice result of improving singular model.
2. The proposed method of using weights as data to learn via a VAE is novel, interesting, and a potential direction to expand upon in future work, especially given interesting results like point 1. Also, it appears that only a few models worth (2 Gemma models) is sufficient to train on to achieve useful results.
3. This method can achieve heterogenous merging, mostly on intra-family based merges.

**Weaknesses:**

1. While the results are impressive, the description of the method is not entirely clear at times and detracts from the contribution of the paper. For example, it is not clear how heterogenous rescaling occurs according to the description, and it is not when the OT based alignment occurs in the latent space training. The contribution of the paper seems solid, but its presentation seems rushed and the paper does not seem reproducible or easily understandable in its current state. Another example is the statement that the evaluation of cross family evaluation is performed using lm-eval due to issues with the llama model when using previous evaluation code. What does this mean here?
2. Despite good results, this work lacks some analysis of what is learned by the VAE model, as well as some ablations of key choices. It does not seem clear why this method works, or what about it makes it work. What is the latent size used in this work? And what is the chunked size? And how were these values set?

This paper is quite interesting but unfortunately its presentation and polish is very lacking, which brings into question the correct execution of this work. I think this paper could be impactful and a nice contribution, but in its current form I cannot recommend accepting it as it seems only partially finished.

**Questions:**

1. In section 3.1, is the v_proj included in this analysis? It is missing from the description of the moment analysis of the key LLM weights.
2. What exactly is a layer matrix in section 3.1 line 150 and section 3.2 line 204? Is it a single weight like q_proj or up_proj or is it the entire Transformer layer?
3. Is the embedding from line 206 part of the transformer encoder? Or is it separate?
4. What is the exact operation for the rescaling procedure described for heterogenous mapping? It is not clear how the value r is used in this mapping.
5. What is the model used as base in Table 5? And which model is weighted 0.1 in the mixture?

Typos
1. Line 192, porjection
2. Line 139 up_porj, down_porj
3. Line 233 artihmetic
4. What is the "fixed of 2" mean on line 321?
5. Line 796 is missing a reference to a figure in Latex.
6. Line 403 familly
7. Line 404 "is perform"

---

> ### Author Response · Authors · 2025-11-19
>
> We thank the reviewer for their careful reading and constructive comments. We address all weaknesses and questions below.
>
> ---
>
> ## Weaknesses
>
> ### Heterogeneous Rescaling / Proportional Mapping
>
> The **Heterogeneous Mapping (Proportional Rescaling)** is implemented **inside the VAE encoder during training** for heterogeneous base architectures (e.g., Gemma-3-4B-it vs. Gemma-3-1B-it). During VAE training, we design the encoder so that per-layer latent representations across different architectures have the **same latent dimensionality** (d).
> While the VAE always maps each layer to a latent of dimension (d), the underlying architectures differ in the **number of layers** ($n_s$ vs. $n_t$) and in the **size** of the corresponding weight tensors ($M$ vs. $N$). We therefore use the ratio $r = \frac{n_t N}{n_s M}$
>
> as a conceptual guide for designing the encoder and its linear projections, so that after encoding we obtain layer-wise mapped latent representations
> $Z^{(\text{src,mapped})}, Z^{(\text{tgt})} \in \mathbb{R}^{n_t \times d}.$
> This ensures that source and target latents are comparable at the layer level and can be merged in a consistent way (as described in Algorithm 1, step 4). This heterogeneous rescaling is only needed when the base architectures differ; for homogeneous setups, a shared encoder suffices without additional rescaling.
>
> ### Optimal Transport (OT) Alignment
>
> **OT alignment** is performed  in the latent space  after VAE encoding (which already includes the heterogeneous rescaling above) and  before any interpolation or merging (Figure 1b, Algorithm 1, step 5).
>
> It is used **only for heterogeneous merges**, i.e., when we combine experts from different architectures or from separately trained VAEs. In this case, even if the latents have the same dimension (d), the two encoders may have learned distinct manifolds, so naive interpolation between source and target latents tends to be unreliable and often degrades performance.
>
> **OT alignment** addresses this by statistically mapping the source latent distribution to the target latent distribution before we inject the source signal into the target model. All heterogeneous merging results that rely on this mechanism are reported in **Tables 4–6** and illustrated in **Figure 4(b)**.
>
> ### Reproducibility and the `lm-eval` Library
>
> We acknowledge that our original description of the evaluation environment was not sufficiently clear.
>
> We initially used the benchmark evaluation scripts from [1] for merging gemma family models. However, when we extended our experiments to heterogeneous merging involving LLaMA-family models, we encountered unstable results for llama-3.2-1b instruct with that legacy evaluation code. These issues stemmed from the older, unmaintained evaluation stack rather than from our LS-Merge method itself.
>
> To ensure robust and reproducible reporting against standard baselines, we moved the evaluation for **Tables 4–6** to the actively maintained **`lm-eval-harness`**. This change is purely infrastructural:
>
> * The evaluation tool does not affect the core behavior of LS-Merge because same issue was raised with the base model with no merging.
> * Our method shows **consistent qualitative behavior and strong performance** regardless of whether we use the original scripts or `lm-eval`.
>
> ### Why the Method Works
> The VAE is trained directly on pretrained LLM weights,  learns a low-dimensional manifold of valid weight configurations where local movements correspond to smooth functional changes in the underlying models.
> The method’s success is twofold:
>
> * **Homogeneous Merging (Tables 2–4).**
>
>   When all experts share the same base architecture, the VAE learns a smooth, continuous latent manifold of weights. Interpolation or sampling within this manifold acts as a form of model steering: by moving along directions that remain inside (or close to) high-performing regions, the method consistently discovers strong checkpoints, often outperforming naive weight-space averaging.
>
> * **Heterogeneous Merging.**
>   For cross-architecture merging, we must overcome both structural and distributional mismatch between models. This is achieved in two steps:
>
>   1. **Dimensional Alignment (Proportional Mapping):**  we enforce a compatible latent parameterization across architectures so that layer-wise latents are comparable.
>
>   2. **Distributional Alignment (OT Alignment):** we align the source and target latent distributions so that interpolation occurs within a shared, meaningful latent geometry rather than across disjoint manifolds.   After these two steps, the source latent is injected **conservatively** (with a small scaling coefficient), so that it behaves like a LoRA-style adapter: it adds complementary capability without destroying the base knowledge of the target model. This controlled latent injection is what enables non-trivial gains in the heterogeneous setting, where naive latent mixing would otherwise lead to severe performance degradation.

---

> ### Author Response · Authors · 2025-11-19
>
> ### VAE Parameters and Ablation Justification
>
> Across all experiments, we use a **fixed chunk size** and **fixed latent dimension**:
>
> * **Chunk size:** (10,240)
> * **Latent dimension per chunk:** (640)
>
> Concretely, each flattened weight chunk of length (10,240) is encoded into a 640-dimensional latent vector.
>
> We chose these values based on **preliminary experiments and ablation studies** (Section 5.2 and Appendix B.1 in the main paper). In these ablations, we varied
> * the **latent dimensionality**,
>
> and evaluated performance on **unseen models** whose initial weights were reconstructed from VAE latents.
> We observed:
>
> * **Too aggressive compression** (very small latent dimension at fixed chunk size) leads to severe performance degradation on unseen models, indicating poor reconstruction and overfitting of the VAE.
> * **Very small chunk sizes** dramatically increase the number of chunks per model, slowing VAE training significantly without providing clear downstream benefits.
>
> The final configuration (chunk size (10,240), latent dimension (640)) provides a moderate bottleneck: it yields stable VAE training, good reconstruction on held-out weights, and competitive performance on all reported benchmarks.
>
> [1] *Model Swarms: Collaborative Search to Adapt LLM Experts via Swarm Intelligence*.
>
> ---
>
> ## Response to  Questions
>
> ---
>
> ### Q1: Is `v_proj` included in the analysis in Sec. 3.1?
>
> **Yes, `v_proj` is included.** The moment and kurtosis analyses cover all four projection matrices of the self-attention layer (`q_proj`, `k_proj`, `v_proj`, and `o_proj`), as well as the three MLP projections (`up_proj`, `down_proj`, `gate_proj`).
>
> The omission of `v_proj` in one sentence of Section 3.1 is a typo. We will correct the sentence to:
>
>  “We compute the first four moments for self-attention projections (`q_proj`, `k_proj`, `v_proj`, `o_proj`) and MLP projections (`up_proj`, `down_proj`, `gate_proj`).”
>
> ---
>
> ### Q2: What exactly is a “layer matrix” in Sec. 3.1/3.2?
>
> By a **“layer matrix”** ($W \in \mathbb{R}^{n \times m}$), we mean a **single 2D weight tensor** (for example, `q_proj` or `up_proj`), not an entire Transformer block. In the text, this term was used somewhat interchangeably with “parameter signal” and “weight tensor,” which may have caused confusion.
>
> In the revised version, we will replace “layer matrix” with the clearer term **“weight tensor”** throughout Sections 3.1–3.2.
>
> ---
>
> ### Q3: Is the embedding in Sec. 3.2 part of the transformer encoder?
>
> **Yes, conceptually the embedding is part of the encoder.** It is implemented as a shared linear projection
> (h: $\mathbb{R}^c \to \mathbb{R}^d$)
> that maps each input weight chunk of length \(c\) (here, (c = 10,240)) to the embedding dimension (d) (here, (d = 640)).
>
> This projection (h) is trained **jointly** with the downstream Transformer encoder ($E_\theta$) and serves as the input layer that converts chunked weights (X) into the embedded token sequence ($X_{\text{emb}}$) processed by the encoder. We will make this explicit in Section 3.2 by stating that (h) is conceptually part of the encoder and is learned end-to-end with ($E_\theta$).
>
> ---
>
> ### Q4: Exact operation for heterogeneous rescaling and use of (r)
>
> The **Proportional Mapping (heterogeneous rescaling)** is used for cross-architecture merges to ensure that the **total latent capacity** devoted to a given layer type matches between the source and target models.
>
> Consider a layer type such as `k_proj`:
>
> * The **source** model has ($n_s$) layers of this type, each with (M) parameters.
> * The **target** model has ($n_t$) layers of this type, each with (N) parameters.
> * The chunk size is fixed to \(c\).
>
> Then the number of chunks for this layer type is
> $\text{chunks}_s = n_s \cdot \frac{M}{c}, \qquad
> \text{chunks}_t = n_t \cdot \frac{N}{c}.$
> Let ($d_s$) and ($d_t$) be the latent dimensions used by the encoder for the source and target, respectively. To make the total latent capacity match (and avoid truncating or dropping layers from the deeper model), we require $n_s \cdot \frac{M}{c} \cdot d_s = n_t \cdot \frac{N}{c} \cdot d_t,$
> which gives $\frac{d_s}{d_t} = \frac{N\cdot n_t}{M\cdot n_s}.$
>
> This ratio defines the proportional mapping factor \(r\). Using this mapping, we choose ($d_s$) and ($d_t$) so that, after encoding and alignment, the latent representations match in dimension: $Z^{(\mathrm{src,mapped})}, Z^{(\mathrm{tgt})} \in \mathbb{R}^{n_t \times d}.$
>
> We will clarify these formulas explicitly in Section 3.3 and reference them from Algorithm 1.
>
> ---
>
> ### Q5: Base model in Table 5 and which model is weighted 0.1
>
> **Concern.** It is not clear what model is used as “Base” in Table 5, and which model corresponds to the weight (0.1) in the mixture.
>
> **Answer.** In Table 5, we perform a cross-family merge between Gemma and LLaMA:
>
> * The **Base model (target)** is **Gemma-3-1B-it**.
> * The **model with weight (0.1) (source)** is **LLaMA-3.2-1B-instruct**, after OT alignment in latent space.

---

> ### Author Response · Authors · 2025-11-19
>
> ### Q-5 continue
>
>
> In other words, OT+interp. injects a small amount of the OT-aligned LLaMA-3.2-1B-instruct latent into Gemma-3-1B-it, with mixing weight ($\lambda = 0.1$). The goal is to add capability from the source model while preserving the target model’s core behavior.
>
> We will update the caption of Table 5 and the text in Section 4.4 to state explicitly:
>
> > “Base denotes Gemma-3-1B-it. OT+interp. interpolates between Gemma-3-1B-it (target) and OT-aligned LLaMA-3.2-1B-instruct (source) with ($\lambda = 0.1$).”
>
> We have also carefully polish the paper  and will include these clarification upon addressing your concerns.
>
> We have carefully polished the paper, checked the typos, and will include these clarifications in the revised version. We hope that our revisions and explanations address your concerns.

---

> > ### Author Response · Authors · 2025-11-28
> >
> > Dear Reviewer,
> >
> > We hope you are doing well.
> > This is a gentle reminder regarding our rebuttal. We understand that reviewers are extremely busy, and we sincerely appreciate the time and effort you dedicate to the evaluation process. We have provided comprehensive responses to your comments and hope they help address your initial concerns.
> >
> > If any further clarification or additional material would assist your assessment, we would be happy to provide it promptly.
> >
> > Best regards,
> > The Author

---

### Official Review · Reviewer_XrKv · 2025-11-04

**Soundness:** 4
**Presentation:** 4
**Contribution:** 4
**Rating:** 8
**Confidence:** 3

**Summary:**

This paper presents a novel method for model merging where the size of the models do not need to match. In order to do so, the authors use variational auto-encoders, optimal transport, linear interpolation,  and projections down to a merged weight space. The results beat strong baselines (including very recently published methods) on a wide variety of tasks. I’ll keep this reviews short because, overall, I would be very happy to see this published in the conference.

**Strengths:**

Though the paper is dense and covers a lot of complex topics, it is well-written and easy to follow. In addition, things are presented clearly, such as the caption of Figure 1.

There are a lot of experimental results with very strong baselines that they beat.

It is an interesting idea intellectually and I would have liked to have seen the paper published even if the results had not beaten the baselines – but they did.

**Weaknesses:**

Perhaps I missed it, but I think a bit more background on OT could be useful for the presentation to the reader. The paragraph at the end of section 3 could be expanded a bit more. I ended up needing to look at one of the cited papers. However, most of the other parts of the paper explained complex topics very well.

**Questions:**

The method seems like it could be a bit computationally expensive - and this is mentioned in the limitations. How expensive is it exactly? I don't have a particular way I'd like to see this question answered, but whatever makes sense. For instance, maybe time on a GPU? What sort of GPU (which is definitely dependent on the models)? Maybe percent of compute needed compared to doing something from scratch? Or comparison to another method (i.e. AIM)?

---

> ### Author Response · Authors · 2025-11-19
>
> We thank the reviewer for their thoughtful and constructive feedback.
>
> ---
>
> In this response, we clarify the role of Optimal Transport (OT) in LS-Merge, provide a more accessible OT background, and give a detailed of the computational resources of our method, including a comparison with AIM. We also address the specific questions regarding architectural details, the definition of “layer matrix,” and the embedding module, as well as minor wording and presentation issues.
>
> ## Brief Expanded Background on Optimal Transport for LS-Merge
>
> ### The alignment challenge
>
> When merging neural networks in latent space, source and target latents may share dimensionality but have mismatched geometries and distributions. Naive interpolation between such misaligned representations can cause semantic mismatches (similar functions far apart), distributional collapse (interpolates off-support), and loss of important geometric structure. Optimal Transport (OT) provides a principled solution by finding the *minimal-cost transformation* that aligns one distribution to another while preserving as much structure as possible.
>
> This is particularly crucial for neural network weights, where the geometry of the parameter space encodes learned features and their relationships.
>
> ### Optimal Transport Core concept
>
> Consider two probability distributions μ and ν on $ℝ^d$ representing, in our context, the latent distributions of source and target models respectively. The optimal transport problem asks, "What is the most efficient way to transform μ into ν?*
>
> **Transport Map**
> A measurable map T: $\mathbb{R}^d → \mathbb{R}^d$ *transports* $\mu$ to $\nu$, denoted $T_{\\\#}\mu = \nu$, if for all Borel sets $ B ⊆ \mathbb{R}^d: \nu(B) = \mu(T^{-1}(B))$
>
> Intuitively, if we sample $X \sim \mu$ and apply T, then $T(X) \sim \nu$.
>
> Among all possible transport maps, we seek the one minimizing the average squared displacement:
>
> $ T* = arg min_{T: T_{\\\#}\mu= ν} ∫_{ℝ^d} ||T(x) - x||^2 dμ(x) \quad   (Monge)$
>
> This quadratic cost $c(x,y) = |x-y|^2$ induces the 2-Wasserstein metric (capturing both geometric and distributional differences), admits a unique optimal map under mild conditions (Brenier's theorem)[9], and yields Wasserstein geodesics that coincide with displacement interpolation.
>
> #### The Kantorovich relaxation
>
> Direct optimization over transport maps is computationally intractable. Kantorovich's insight was to relax the problem by considering *transport plans*—joint distributions π on ℝ^d × ℝ^d with marginals μ and ν:
>
> $\Pi* = arg min_{π ∈ Π(μ,ν)} ∫_{ℝ^d × ℝ^d} ||x-y||^2 dπ(x,y) \quad (Kantorovich) $
>
> where $\Pi(μ,ν)$ denotes the set of all couplings with correct marginals.
>
> From plans to maps: When μ is absolutely continuous, the optimal plan $\Pi*$ is supported on the graph of a unique transport map $T*$, recoverable via: $T*(x) = ∇φ(x)$ for some convex potential $φ$. This connection is crucial for our implementation.
>
> ---
>
> #### Algorithm: OT-based Distributional Alignment
>
> ---
> #### Input and Output
> * **Input:**
>     * Source latents $Z_{src}= \\{z_{i}^{src}\\}_\{i=1\}^n$,
>     * Target latents $Z_{tgt} = \\{z_{j}^{tgt}\\}_\{j=1\}^n$
>
>
> * **Output:** Aligned source latents $\tilde{Z}_{src}$
>
> #### Procedure
>
> 1.  **Step 1: Construct empirical measures (equal mass)**
>     * $\mu_{\text{src}} \leftarrow \frac{1}{n} \sum_{i=1}^{n} \delta_{z_i^{\text{src}}}$
>     * $\mu_{\text{tgt}} \leftarrow \frac{1}{n} \sum_{j=1}^{n} \delta_{z_j^{\text{tgt}}}$
>
> 2.  **Step 2: Solve discrete Optimal Transport (OT) problem**
>     * $ \Gamma^\star \leftarrow solve(\mu_{src}, \mu_{tgt})$ // *Using POT library*
>
> 3.  **Step 3: Construct approximate transport map via barycentric projection**
>     * **FOR** each $z_i^{src} \in Z_{src}$:
>
>         $\tilde z_i \leftarrow \frac{\sum_j \gamma_{ij}^\star z_j^{\text{tgt}}}{\sum_j \gamma_{ij}^\star}$  *(weighted barycenter)*
>
>     * **ENDFOR**
>
>   **Return**   $\\tilde Z\_{\text{src}} = \\{\\tilde z\_i\\}\_{i=1}^{n}$
>
>
> ---
>
> ### Application of OT in LS-Merge
>
> LS-Merge operationalizes the above OT framework through the following algorithmic 1 pipeline in the paper. Importantly, we sample the same number of latent vectors n from both source and target models, ensuring balanced representation and simplifying the transport problem.
> After OT alignment, we perform the actual model merging in latent space.
>
> ---
> ### Theoretical Justification and Practical Benefits
>
> The OT-aligned interpolation can be viewed as an approximation to the Wasserstein geodesic between $μ_src$ and $μ_tgt$. This has minimal distortion, structure preservation, avoids mode collapse.

---

> ### Author Response · Authors · 2025-11-19
>
> #### Computational considerations
>
> For typical layer sizes in our experiments (n ~ 10³), the discrete OT problem involves an n × n transport plan. This square structure can be solved efficiently using Sinkhorn iterations with entropic regularization: O(n²/ε²) for ε-approximation.
>
> We use the POT library [4] which provides optimized implementations of these algorithms, with automatic selection based on problem size.
>
>
> ### References
>
> [1] C. Villani, *Optimal Transport: Old and New*. Springer, 2009.
>
> [2] F. Santambrogio, *Optimal Transport for Applied Mathematicians*. Birkhäuser, 2015.
>
> [3] L. Ambrosio, N. Gigli, and G. Savaré, *Gradient Flows in Metric Spaces and in the Space of Probability Measures*. Birkhäuser, 2008.
>
> [4] G. Peyré and M. Cuturi, "Computational Optimal Transport," *Foundations and Trends in Machine Learning*, 11(5–6):355–607, 2019.
>
> [5] R. Flamary, N. Courty, et al., "POT: Python Optimal Transport," *Journal of Machine Learning Research*, 22(78):1–8, 2021.
>
> [6] M. Cuturi, "Sinkhorn Distances: Lightspeed Computation of Optimal Transport," *NeurIPS*, 2013.
>
> [7] N. Courty, R. Flamary, et al., "Optimal Transport for Domain Adaptation," *IEEE Trans. Pattern Analysis and Machine Intelligence*, 39(9):1853–1865, 2017.
>
> [8] A. Genevay, G. Peyré, and M. Cuturi, "Learning Generative Models with Sinkhorn Divergences," *AISTATS*, 2018.
>
> [9] Maggi F. The Brenier Theorem. In: Optimal Mass Transport on Euclidean Spaces. *Cambridge Studies in Advanced Mathematics*. Cambridge University Press; 2023:59-66.
>
> we will provide proper description in the paper
>
>
> How expensive is the method computationally?
>
> The main additional computation occurs during training the VAE encoder–decoder, which is performed once per architecture family and then reused for all subsequent merges. In our experiments, this pretraining step takes roughly 30 minutes to 2 hours on a single GPU, depending on the architecture size and available compute. Once the VAE is trained, each merge itself is lightweight, involving only encoding, OT-based alignment, and decoding in latent space. Since our approach is inherently learning-based, this one-time VAE training cost is analogous to the auxiliary training overhead present in many learning-based methods and is amortized over many merges, rather than being a recurring computational burden specific to our approach.
>
> | Aspect                          | **LS-Merge (Ours)**                                           | **AIM**                                                       |
> |---------------------------------|---------------------------------------------------------------|----------------------------------------------------------------|
> | Uses training data for merge    | **No** – operates only on pretrained weights                 | **Yes** – requires a dataset to compute activations           |
> | Forward passes through LLM      | **None**                                                      | **Required** (to collect activations)                             |
> | Backward / optimization steps   | **None**                                                      | **None**                         |
> | Main compute domain             | Latent weight space (encode → OT align → decode)             | Activation space + parameter updates                          |
> | Cost scaling                    | One-time, dataset-free; scales with #layers and model size   | Scales with dataset size, sequence length, and #iterations    |
> | Output after merge              | Multiple standard checkpoints                                   | Single standard checkpoint                                    |
> | Inference-time architecture     | Unchanged (same as base/target model)                        | Unchanged (same as base/target model)                         |
> | Per-token inference FLOPs       | Same as underlying LLM                                       | Same as underlying LLM                                       |
> | Extra modules at inference      | None                                                          | None                                                          |
>
> Our method is dataset-independent and requires no calibration data at inference. In contrast, AIM must run each LLM on a calibration set, which is significantly more expensive than our VAE-based inference plus OT computed on only 20 latent samples. Our approach also yields multiple merged checkpoints rather than a single one, although weight-space merging methods such as DARe-TiES remain faster than ours at inference time.
>
> we plan to add additional ablation study showing the computation cost
>
> We hope that these clarifications make the method easier to follow, demonstrate that the computational overhead is modest and amortized, and address the concerns about readability and reproducibility.

---

### Author Response · Authors · 2025-12-03

We thank the reviewers and the Area Chair for evaluating our work. We have provided detailed responses to all reviewer comments and revised the paper accordingly.

**OT Alignment (Section 3.3):** We substantially rewrote this section to directly address the reviewers' concerns regarding clarity and motivation. We also added Figure 9 and Section C in the appendix to visually illustrate why alignment is necessary for heterogeneous model merging. Additionally, we provide further details regarding the training process in the appendix.

**Training Time and Practical Considerations:** We expanded the appendix with a new paragraph describing training time, runtime characteristics, and strategies for speeding up alignment in heterogeneous settings.

**Additional Ablation Study:** In the main paper, we added a new ablation comparing our VAE to PCA, highlighting the clear advantage of non-linear manifold learning over linear or incremental PCA for functional weight reconstruction.

Additional evaluations were conducted using the latest version of `lm-evaluation-harness`.

we have uploaded the revised copy of the paper.
We believe these revisions significantly strengthen the clarity, completeness, and empirical justification of the method. We appreciate the Area Chair's consideration.

---

### Meta-Review · Area_Chair_CthG · 2026-01-07

**Summary:**

The paper proposes LS-Merge, a method for merging language models in a learned latent space using a transformer-based VAE, enabling heterogeneous model merging (e.g., different architectures or sizes). The approach involves encoding weight chunks into latents, aligning distributions via Optimal Transport (OT) for cross-architecture cases, interpolating in latent space, and decoding back to weights. Reviewers generally agree the idea is novel and well-motivated, with strong empirical results across multiple settings (self-merging, LoRA fusion, cross-family merging). However, concerns were raised regarding presentation clarity (Reviewer 593M), lack of analysis on VAE reconstruction fidelity and failure modes (Reviewer mkDJ), and computational cost (Reviewer XrKv). The authors provided detailed rebuttals addressing all points, including clarifications on OT alignment, heterogeneous rescaling, VAE hyperparameters, reproducibility, and added ablations (e.g., vs. PCA).

**Reviewer Concerns:**

- Reviewer XrKv: The computational cost question was thoroughly answered—VAE training is one-time (~30min–2h/GPU), merge inference is lightweight, and no calibration data is needed (unlike AIM). The OT background was also clarified with expanded explanation and references.
- Reviewer 593M: All technical ambiguities (e.g., “layer matrix,” heterogeneous rescaling, OT timing, base model in Table 5) were responded with precise definitions and algorithmic details. Typos and presentation issues were acknowledged and will be corrected. The lm-eval switch was justified as an infrastructural fix for reproducibility.
- Reviewer mkDJ: The link between VAE compression ratio and downstream performance was confirmed via existing ablations (Section 5.2, Table 7); additional KL-weight ablation will be added. Failure conditions were explained: poor VAE generalization under high compression leads to degraded merges.

**Reviewer Scores:**

Reviewer XrKv: Originally scored 8. The rebuttal addressed the only weakness (OT background and compute cost). Likely would maintain at 8.

Reviewer 593M: Originally scored 4, citing poor presentation and lack of clarity. The rebuttal comprehensively resolves all ambiguities and commits to textual improvements. Given the solid contribution and now-clear methodology, the score would likely rise to 6.

Reviewer mkDJ: Originally scored 8. The rebuttal adequately addresses both weaknesses with empirical evidence and planned ablations. No reason to lower score; likely maintains 8.

---

### Decision · Program_Chairs · 2026-01-26

Accept (Poster)